# Feeding state-dependent regulation of developmental plasticity via CaMKI and neuroendocrine signaling

Scott J Neal[1], Asuka Takeishi[1], Michael P O'Donnell[1], JiSoo Park[2], Myeongjin Hong[2], Rebecca A Butcher[3], Kyuhyung Kim[2]*, Piali Sengupta[1]*

[1]Department of Biology, National Center for Behavioral Genomics, Brandeis University, Waltham, United States; [2]Department of Brain and Cognitive Sciences, Daegu Gyeongbuk Institute of Science and Technology (DGIST), Daegu, Republic of Korea; [3]Department of Chemistry, University of Florida, Gainesville, United States

**Abstract** Information about nutrient availability is assessed via largely unknown mechanisms to drive developmental decisions, including the choice of *Caenorhabditis elegans* larvae to enter into the reproductive cycle or the dauer stage. In this study, we show that CMK-1 CaMKI regulates the dauer decision as a function of feeding state. CMK-1 acts cell-autonomously in the ASI, and non cell-autonomously in the AWC, sensory neurons to regulate expression of the growth promoting *daf-7* TGF-β and *daf-28* insulin-like peptide (ILP) genes, respectively. Feeding state regulates dynamic subcellular localization of CMK-1, and CMK-1-dependent expression of anti-dauer ILP genes, in AWC. A food-regulated balance between anti-dauer ILP signals from AWC and pro-dauer signals regulates neuroendocrine signaling and dauer entry; disruption of this balance in *cmk-1* mutants drives inappropriate dauer formation under well-fed conditions. These results identify mechanisms by which nutrient information is integrated in a small neuronal network to modulate neuroendocrine signaling and developmental plasticity.

*For correspondence: khkim@ dgist.ac.kr (KK); (KK); sengupta@ brandeis.edu (PS) (PS)

**Competing interests:** The authors declare that no competing interests exist.

## Introduction

Discrete alternate phenotypes arising from a single genotype in response to varying environmental cues is referred to as polyphenism (*Michener, 1961*; *Mayr, 1963*; *Stearns, 1989*). Well-described polyphenic traits include the exhibition of wings on locusts, caste hierarchy in social insects, and environmental sex determination in reptiles (*Nijhout, 2003*; *Beldade et al., 2011*; *Simpson et al., 2011*). In well-studied cases as in insects, it has been shown that animals integrate sensory cues during specific developmental stages to promote the expression of alternate phenotypic traits via regulation of endocrine and neuromodulatory signaling (*Simpson et al., 2011*; *Watanabe et al., 2014*). Environmental cues that trigger developmental plasticity include pheromones, temperature, mechanical stimuli, and food (*Nijhout, 2003*; *Simpson et al., 2011*). In particular, nutrient availability and quality during development is a major regulator of polyphenism in many species (*Wheeler, 1986*; *Greene, 1989*; *Pfennig, 1992*; *Bento et al., 2010*). Although the extent and adaptive value of polyphenism has been extensively discussed (*Pfennig et al., 2010*; *Moczek et al., 2011*), the underlying molecular and neuronal mechanisms that allow animals to sense and integrate signals from food and feeding-state signals in the context of other cues to regulate phenotypic plasticity are not fully understood.

*Caenorhabditis elegans* exhibits polyphenism in response to environmental cues sensed during a critical period in development. Shortly following hatching, *C. elegans* larvae assess crowding in their environment via concentrations of a complex mixture of small molecules called ascarosides

**eLife digest** Living organisms have the remarkable ability to adapt to changes in their external environment. For example, when conditions are favorable, the larvae of the tiny roundworm *C. elegans* rapidly mature into adults and reproduce. However, when faced with starvation, over-crowding or other adverse conditions, they can stop growing and enter a type of stasis called the dauer stage, which enables them to survive in harsh conditions for extended periods of time. The worms enter the dauer stage if they detect high levels of a pheromone mixture that is produced by other worms—which indicates that the local population is over-crowded. However, temperature, food availability, and other environmental cues also influence this decision.

A protein called TGF-β and other proteins called insulin-like peptides are produced by a group of sensory neurons in the worm's head. These proteins usually promote the growth of the worms by increasing the production of particular steroid hormones. However, high levels of the pheromone mixture, an inadequate supply of food and other adverse conditions decrease the expression of the genes that encode these proteins, which allows the worm to enter the dauer state. It is not clear how the worm senses food, nor how this is integrated with the information provided by the pheromones to influence this decision.

To address these questions, Neal et al. studied a variety of mutant worms that lacked proteins involved in different aspects of food sensing. The experiments show that worms missing a protein called CaMKI enter the dauer state even under conditions in which food is plentiful and normal worms continue to grow. CaMKI inhibits entry into the dauer stage by increasing the expression of the genes that encode TGF-β and the insulin-like peptides in sensory neurons in response to food.

Neal et al.'s findings reveal how CaMKI enables information about food availability to be integrated with other environmental cues to influence whether young worms enter the dauer state. Understanding how food sensing is linked to changes in hormone levels will help us appreciate why and how the availability of food has complex effects on animal biology and behavior.

(collectively referred to as dauer pheromone) produced by conspecifics (*Golden and Riddle, 1982*, *1984c*; *Jeong et al., 2005*; *Butcher et al., 2007*; *Edison, 2009*; *Ludewig and Schroeder, 2013*). High concentrations of one or more of these chemicals promote entry of larvae into the alternate stress-resistant and long-lived dauer developmental stage, whereas under uncrowded conditions, larvae continue in the reproductive cycle (*Cassada and Russell, 1975*) (*Figure 1A*). Although phero-mone is the instructive cue for dauer entry, additional cues, such as temperature and food availabil-ity, also regulate this binary decision (*Golden and Riddle, 1984a*, *1984b*, *1984c*; *Ailion and Thomas, 2000*) (*Figure 1A*). Thus, high (low) concentrations of food or low (high) temperature can efficiently inhibit (promote) pheromone-induced dauer formation, allowing animals to assess and integrate diverse sensory cues in order to make a robust developmental choice.

Decades of investigation have shown that environmental stimuli detected by sensory neurons modulate neuroendocrine signaling to regulate the choice of larval developmental trajectory in *C. elegans* (*Fielenbach and Antebi, 2008*). High pheromone concentrations, low food abundance and high temperature cues downregulate expression of the *daf-7* TGF-β ligand and several insulin-like peptide (ILP) genes in subsets of ciliated sensory neurons in the head amphid organs (*Ren et al., 1996*; *Schackwitz et al., 1996*; *Li et al., 2003*; *Cornils et al., 2011*; *Entchev et al., 2015*) (*Figure 1A*). Downregulated TGF-β and insulin signaling in turn decrease biosynthesis of dafachronic acid steroid hormones by neuronal and non-neuronal endocrine cells (*Fielenbach and Antebi, 2008*). In the absence of these steroid hormones, the DAF-12 nuclear hormone receptor promotes dauer entry, whereas in the ligand-bound form, DAF-12 promotes reproductive development (*Antebi et al., 1998*, *2000*; *Ludewig et al., 2004*). Ciliated chemosensory neurons required to sense a subset of ascarosides for the regulation of dauer entry have been identified (*Schackwitz et al., 1996*; *Kim et al., 2009*; *McGrath et al., 2011*; *Park et al., 2012*). However, little is known about how food is sensed, and how food signals are integrated with pheromone signals at the level of endocrine gene expression to influence the dauer decision.

Here, we identify the CMK-1 calcium/calmodulin-dependent protein kinase I (CaMKI) as a key player in the regulation of dauer formation as a function of feeding state. Expression of the *daf-7* TGF-β and *daf-28* ILP genes are downregulated in well-fed *cmk-1* mutants, and we find that CMK-1 acts cell-autonomously in the ASI sensory neurons, and non cell-autonomously in the AWC sensory neurons, to regulate the expression of *daf-7* and *daf-28*, respectively. We show that the subcellular localization of CMK-1 in AWC is feeding state-dependent, and that CMK-1 promotes expression of anti-dauer ILP genes in AWC. Our results indicate that a balance of CMK-1-regulated anti-dauer signals from AWC, as well as pro-dauer signals, regulates dauer entry as a function of feeding state, and that this balance is disrupted in *cmk-1* mutants to inappropriately promote dauer entry under well-fed conditions. We also find that basal activity levels in AWC are enhanced upon prolonged starvation in wild-type animals and in well-fed *cmk-1* mutants, and that increased activity acts in parallel with CMK-1 to antagonize dauer formation. Together, these results identify CMK-1 CaMKI as a key molecule that encodes information about nutrient availability within a sensory neuron network to regulate neuroendocrine signaling and a polyphenic developmental choice.

## Results

### *cmk-1* mutants form dauers inappropriately in the presence of pheromone and food

To verify that pheromone-induced dauer formation in wild-type animals is suppressed by bacterial food, we quantified dauers formed in the presence of pheromone and different concentrations of non-replicative (heat-killed) as well as replicative (live) bacteria. Although no dauers were observed in the absence of added pheromone (*Figure 1—figure supplement 1A*; *Figure 1—source data 2*), >80% of wild-type larvae entered into the dauer stage on plates containing 6 μM ascr#3 (also referred to as asc-ΔC9 or C9) and l60 μg heat-killed OP50 bacteria (*Figure 1B*; *Figure 1—source data 1*). Dauer formation decreased upon increasing the amount of heat-killed bacteria and was fully suppressed by only 80 μg of live bacteria (*Figure 1B*). We could not reliably quantify the effects of lower concentrations of live bacteria since as reported previously, animals arrest development post-embryonically when food becomes limiting (*Hong et al., 1998*; *Fukuyama et al., 2006*; *Baugh, 2013*). Food also inhibited dauer formation in *daf-22* mutants that are unable to produce most ascarosides (*Golden and Riddle, 1985*; *Butcher et al., 2009*) (*Figure 1—figure supplement 1B*), indicating that under these conditions, dauer formation is not increased simply due to enhanced endogenous pheromone signaling. These observations confirm that food cues modulate pheromone-induced dauer formation.

To begin to explore the mechanisms by which food signals are integrated with pheromone cues to regulate dauer entry, we focused on genes previously implicated in different aspects of nutrient sensing and/or metabolism in *C. elegans*. These include the *aak-1* and *aak-2* AMP-activated protein kinases (*Apfeld et al., 2004*; *Narbonne and Roy, 2009*; *Cunningham et al., 2012*), the *crh-1* CREB transcription factor and the *cmk-1* CaMKI kinase (*Kimura et al., 2002*; *Suo et al., 2006*; *Suo and Ishiura, 2013*), the *egl-4* cGMP-dependent protein kinase (*Daniels et al., 2000*; *You et al., 2008*), the *eat-4* glutamate transporter (*Avery, 1993*; *Hills et al., 2004*), the *tph-1* tryptophan hydroxylase and *cat-2* tyrosine hydroxylase enzymes (*Sawin et al., 2000*; *Hills et al., 2004*; *Suo et al., 2009*; *Ezcurra et al., 2011*; *Entchev et al., 2015*), and the *skn-1*, *hlh-30* and *mxl-3* transcription factors (*Paek et al., 2012*; *O'Rourke and Ruvkun, 2013*) (*Figure 1C*). Specifically, we reasoned that mutations in genes essential for food signal integration would result in inappropriate entry into the dauer stage in the presence of pheromone and plentiful food but would not lead to constitutive dauer formation (dauer-constitutive or Daf-c) (*Hu, 2007*).

Mutations in the *cmk-1* CaMKI gene fulfilled both these criteria. *cmk-1(oy20)* missense, as well as *cmk-1(oy21)* putative null, mutants consistently formed dauers in the presence of 80 μg live bacteria and 6 μM ascr#3—conditions that fully suppressed dauer formation in wild-type animals (*Figure 1C*). Moreover, few dauers were observed in the absence of added pheromone (*Figure 1—figure supplement 1A*), indicating that *cmk-1* mutants do not form dauers constitutively. *cmk-1* mutants also formed dauers in the presence of live food and ascr#2 (also referred to as asc-C6-MK or C6) and icas#9 (also referred to as IC-asc-C5 or C5) (*Figure 1—figure supplement 1C,D*), indicating that the response was not specific to a particular ascaroside. We could not reliably examine the effects of

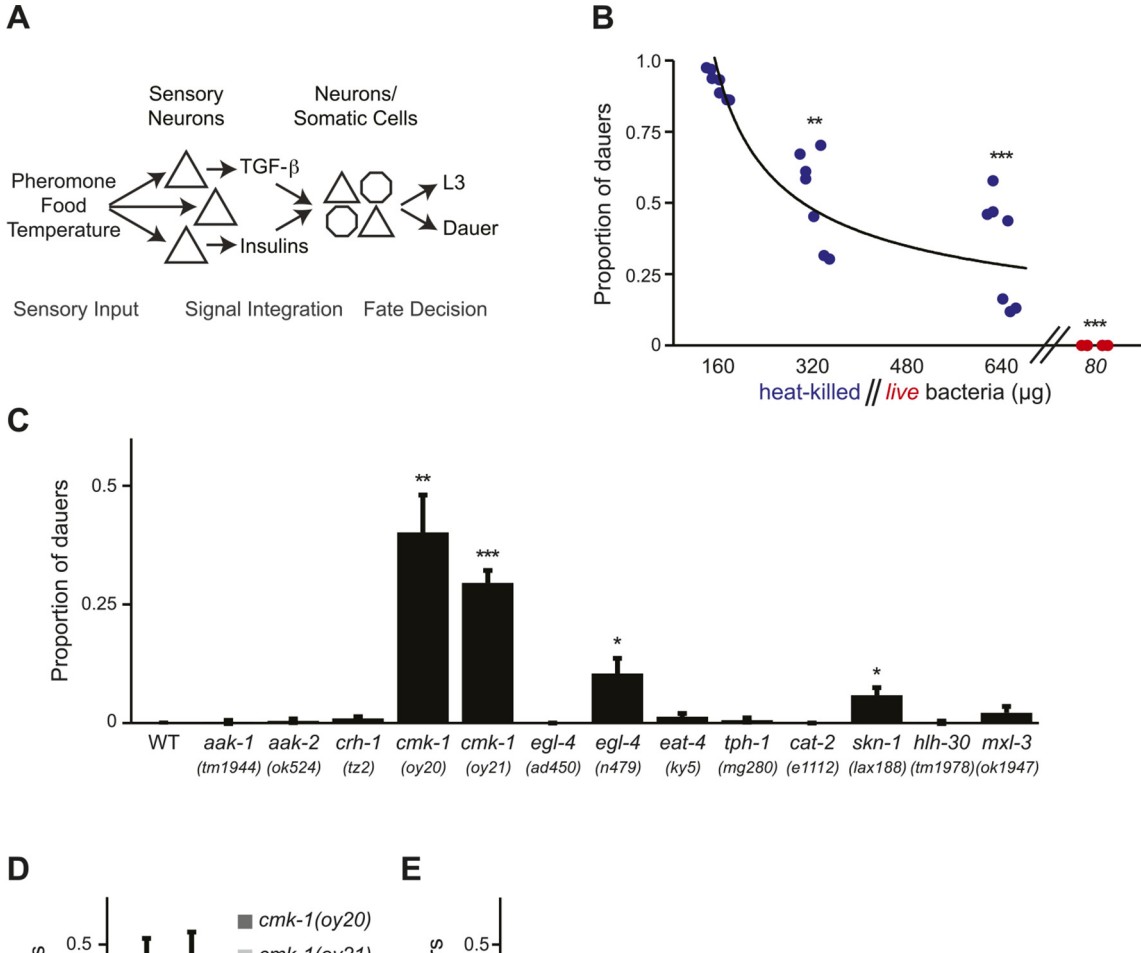

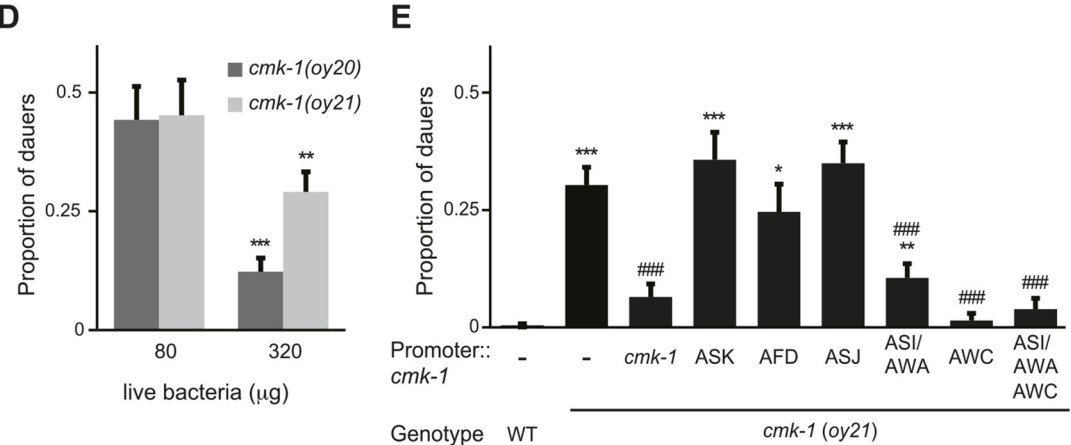

**Figure 1.** CMK-1 acts in the AWC and ASI/AWA neurons to inhibit dauer formation in fed animals. (**A**) Simplified model of sensory inputs modulating TGF-β and insulin signaling in the regulation of the dauer decision. See text for details. (**B**) Quantification of dauer formation in wild-type animals in the presence of 6 μM ascr#3 and the indicated amounts of heat-killed (blue circles) or live (red circles) OP50 bacteria. Each filled circle represents one assay; n > 65 animals per assay, three independent experiments. Line represents best fit to the data. ** and *** indicate different from values using 160 μg of heat-killed bacteria at p < 0.01 and 0.001, respectively (ANOVA and Games-Howell post-hoc test). (**C**) Dauers formed by strains of the indicated genotypes in the presence of 6 μM ascr#3 and 80 μg live OP50. Each data point is the average of ≥3 independent experiments of >65 animals each. Errors are SEM. *, **, and *** indicate different from wild-type at p < 0.05, 0.01, and 0.001, respectively (ANOVA and Games-Howell post-hoc test). (**D**) Dauer formation in *cmk-1* mutants grown with 6 μM ascr#3 and the indicated amounts of live OP50. Each data point is the average of ≥3 independent experiments of >65 animals each. Errors are SEM. ** and *** indicate different from corresponding values using 80 μg OP50 at p < 0.01 and 0.001, respectively (Student's *t*-test). (**E**) Dauers formed by strains of the indicated genotypes grown on plates containing 6 μM ascr#3 and 80 μg live OP50. Promoters used to drive wild-type *cmk-1* cDNA expression were: *cmk-1*p—*cmk-1* upstream regulatory sequences; ASK—*sra-9*p; AFD—*ttx-1*p; ASJ—*trx-1*p; ASI/AWA—*gpa-4*p; AWC—*ceh-36Δ*p. Each data point is the average of ≥3 independent experiments of >65 animals each. For transgenic strains, data are averaged from 1–4 independent lines each. Errors are SEM. *, **, and *** indicate different from wild-type at p < 0.05, 0.01, and 0.001, respectively; ### indicates different from *cmk-1(oy21)* at p < 0.001 (ANOVA and Games-Howell post-hoc test).

*Figure 1 continued on next page*

*Figure 1 continued*

The following source data and figure supplements are available for figure 1:

**Source data 1.** Dauer assay data for individual trials in *Figure 1*.
**Source data 2.** Dauer assay data for individual trials in *Figure 1—figure supplement 1*.
**Figure supplement 1.** CMK-1 inhibits dauer formation in fed animals.
**Figure supplement 2.** *cmk-1p::gfp* is expressed broadly in multiple neurons.

heat-killed bacteria in this assay since *cmk-1* mutants exited the dauer stage prematurely under these conditions (not shown), possibly due to additional metabolic defects that we have not explored further in this study. Increasing the amount of live food to 320 µg decreased but did not fully suppress dauer formation in *cmk-1* mutants (*Figure 1D*), implying that these animals are able to respond to food, but exhibit a shifted threshold of response to feeding. Consistent with the notion that *cmk-1* mutants retain the ability to respond to food cues, egg-laying was modulated by bacterial food in both wild-type and *cmk-1(oy21)* adult animals (*Figure 1—figure supplement 1E*). In addition to food and pheromone, temperature also regulates dauer formation (*Golden and Riddle, 1984b*), and we and others previously showed that *cmk-1* mutants exhibit altered thermosensory behaviors (*Satterlee et al., 2004*; *Schild et al., 2014*; *Yu et al., 2014*). *cmk-1(oy21)* mutants retained the ability to respond to temperature in the context of dauer formation, since dauer formation was fully suppressed at a lower temperature of 20°C in the presence of pheromone (*Figure 1—figure supplement 1F*). We infer that *cmk-1* mutants are defective in correctly integrating food signals into the dauer decision pathway.

## CMK-1 acts in the AWC and ASI/AWA sensory neurons to regulate dauer formation

*cmk-1* is expressed broadly in multiple sensory and non-sensory neuron types in *C. elegans* (*Kimura et al., 2002*; *Satterlee et al., 2004*) (*Figure 1—figure supplement 2*). We first verified that expression of a *cmk-1* cDNA under its endogenous regulatory sequences rescues the dauer formation phenotype in the presence of live bacteria and exogenous ascr#3 (*Figure 1E*). We next performed cell-specific rescue experiments to identify the site(s) of CMK-1 function in the regulation of dauer entry under these conditions. Expression in the ASK pheromone-sensing, or the AFD thermosensory, neurons did not affect the dauer formation phenotype of *cmk-1* mutants (*Figure 1E*), suggesting, but not proving, that CMK-1 does not act simply by modulating pheromone or temperature sensitivity. However, expression of wild-type *cmk-1* sequences in the ASI/AWA or AWC sensory neurons resulted in partial, and nearly complete, suppression of dauer formation, respectively. Expression of *cmk-1* in both ASI/AWA and AWC suppressed dauer formation to the same extent as expression in AWC alone (*Figure 1E*). No rescue was observed upon expression in the ASJ sensory neurons which have also been previously implicated in the regulation of dauer formation (*Bargmann and Horvitz, 1991*; *Schackwitz et al., 1996*) (*Figure 1E*). Thus, CMK-1 acts primarily in AWC but also in ASI/AWA to regulate dauer formation.

## CMK-1 acts in both the TGF-β and insulin pathways to regulate dauer formation

The DAF-7 TGF-β and ILP neuroendocrine signaling pathways act in parallel to regulate dauer formation (*Fielenbach and Antebi, 2008*). We asked whether CMK-1 acts in either or both these pathways to regulate dauer formation. *daf-7* TGF-β null mutants are strongly Daf-c, and this phenotype is suppressed upon loss of DAF-3 SMAD or DAF-5 transcription factor function (*Golden and Riddle, 1984c*; *Vowels and Thomas, 1992*; *Thomas et al., 1993*). Despite the presence of multiple ILPs, the *C. elegans* genome encodes a single insulin receptor encoded by the *daf-2* gene. Although loss of function of single ILP genes such as *daf-28* results in only weak effects on dauer formation, likely due to redundancy (*Li et al., 2003*; *Cornils et al., 2011*; *Ritter et al., 2013*; *Hung et al., 2014*), *daf-2*

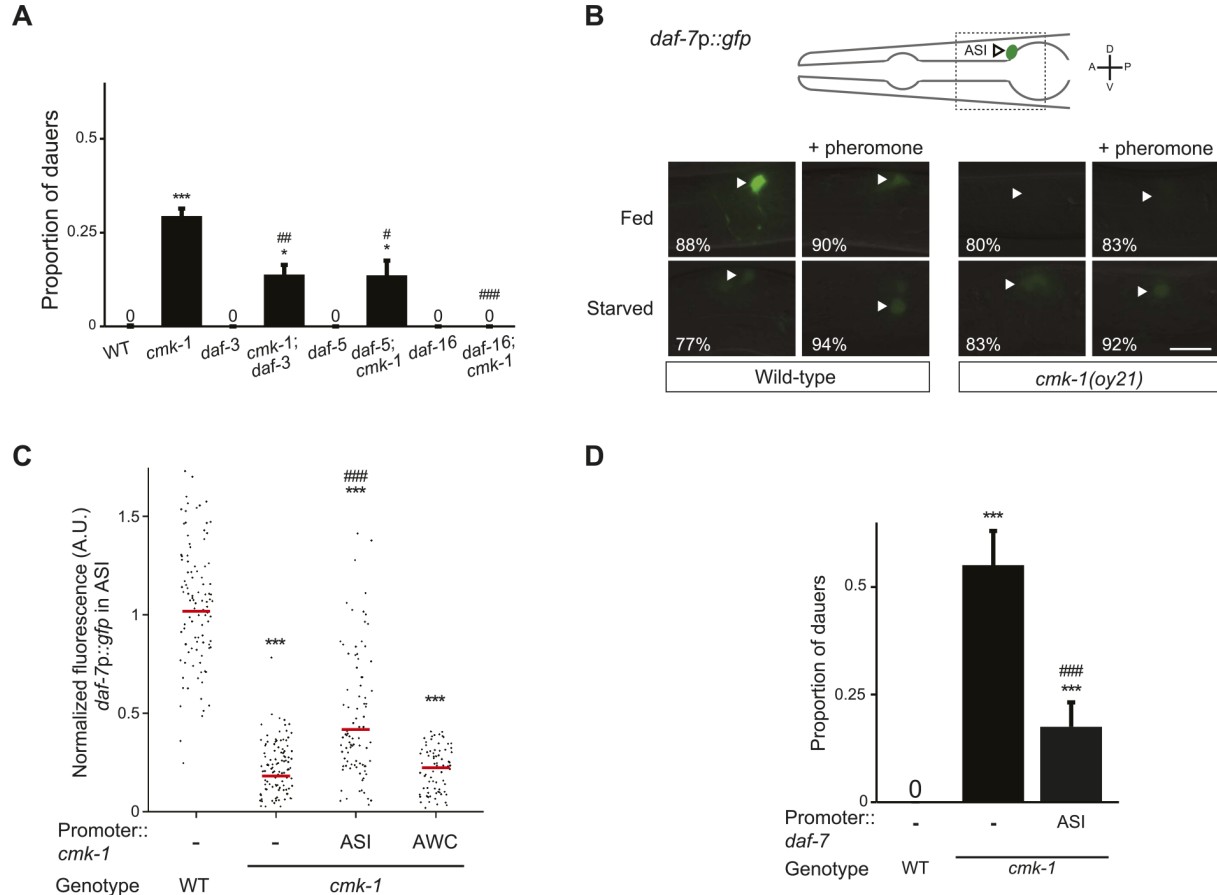

**Figure 2.** CMK-1 acts cell-autonomously to regulate *daf-7* TGF-β expression in ASI. (**A**) Dauers formed by the indicated strains on 80 μg live OP50 and 6 μM ascr#3. Alleles used were: *cmk-1(oy21)*, *daf-3(mgDf90)*, *daf-5(e1385)*, and *daf-16(mgDf50)*. Shown are the averages of ≥3 independent experiments with >65 animals each. Errors are SEM. (**B**) Representative images of *daf-7p::gfp* expression in L1 larvae of wild-type or *cmk-1(oy21)* animals under the indicated conditions. Schematic at top indicates the position of ASI cell body (lateral view); boxed region is shown in panels below. Occasional weak expression is observed in the ADL neurons. Animals were grown with plentiful live OP50 or starved for at least 6 hr in the absence or presence of 1 unit of crude pheromone (see 'Materials and methods'). White arrowheads indicate cell bodies of ASI. Numbers in bottom left hand corners indicate the percentage of examined larvae that exhibit the shown phenotype; n > 50 each; three independent experiments. Lateral view; scale bar: 10 μm. (**C**) Scatter plot of fluorescence intensity of *daf-7p::gfp* expression in ASI in wild-type or *cmk-1(oy21)* mutants. Median is indicated by a red horizontal line. Animals were grown on ample live OP50 in the absence of exogenous pheromone. Each dot is the fluorescence intensity of a single neuron in a given experiment; n > 60 neurons total each, at least three independent experiments. Promoters driving wild-type *cmk-1* cDNA were: ASI—*srg-47*p; AWC—*ceh-36*Δp. (**D**) Dauers formed by shown strains on 80 μg live OP50 and 6 μM ascr#3. The *srg-47* promoter was used to drive expression of wild-type *daf-7* cDNA in ASI. Shown are the averages of ≥3 independent experiments with >65 animals each. Errors are SEM. Unless indicated otherwise, * and *** indicate different from wild-type at p < 0.05 and p < 0.001, respectively, #, ##, and ### indicate different from *cmk-1* at p < 0.05, 0.01, and 0.001, respectively. (ANOVA and Games-Howell post-hoc test).

The following source data is available for figure 2:

**Source data 1.** Dauer assay data for individual trials in *Figure 2*.

insulin receptor mutants are Daf-c (*Thomas et al., 1993*; *Gottlieb and Ruvkun, 1994*; *Gems et al., 1998*). The Daf-c phenotype of *daf-2* mutants is suppressed by loss of *daf-16* FOXO transcription factor function (*Riddle et al., 1981*; *Gottlieb and Ruvkun, 1994*). To determine whether CMK-1 reports food information to either, or both, the TGF-β and insulin pathways, we examined whether *daf-3*, *daf-5*, or *daf-16* mutations suppress the dauer formation phenotype of *cmk-1* mutants. We found that mutations in *daf-3* and *daf-5* partly suppressed the dauer formation defects of *cmk-1* mutants, whereas loss of *daf-16* function fully suppressed this phenotype (*Figure 2A*; *Figure 2—*

*source data 1*). These results suggest that CMK-1 influences both the TGF-β and insulin pathways to regulate dauer formation.

## CMK-1 acts cell-autonomously in ASI, and non cell-autonomously in AWC, to regulate TGF-β and insulin signaling, respectively

Previous work has shown that food and pheromone cues regulate the expression of both *daf-7* TGF-β and the *daf-28* ILP genes to modulate entry into the dauer stage. *daf-7* expression in ASI, and *daf-28* expression in both ASJ and ASI, are downregulated upon starvation or upon exposure to high pheromone concentrations (*Ren et al., 1996*; *Schackwitz et al., 1996*; *Li et al., 2003*; *Entchev et al., 2015*). Since CMK-1 acts in both the TGF-β and insulin pathways to regulate dauer formation, we asked whether the inability of food to fully suppress dauer formation in *cmk-1* mutants is in part due to defects in regulation of expression of one or both ligands in *cmk-1* mutants.

Since the *cmk-1* mutant dauer phenotype is suppressed by downstream mutations in the TGF-β pathway (*Figure 2A*), we first asked whether CMK-1 acts cell-autonomously in ASI to regulate expression of the *daf-7* TGF-β ligand. As reported previously, a *daf-7*p::*gfp* reporter gene was expressed strongly and specifically in the ASI neurons of fed wild-type L1 larvae grown under the conditions of low endogenous pheromone concentrations (*Ren et al., 1996*; *Schackwitz et al., 1996*; *Entchev et al., 2015*) (*Figure 2B,C*). This reporter has been validated to reflect expression of the endogenous *daf-7* gene (*Ren et al., 1996*); the expression level of this gene is one of the components encoding food levels and is variable at all examined food concentrations (*Meisel et al., 2014*; *Entchev et al., 2015*). Expression of *daf-7*p::*gfp* was strongly decreased upon starvation, or in the presence of crude pheromone (*Figure 2B*). We found that unlike in wild-type animals, expression of *daf-7*p::*gfp* was reduced, but not abolished, in *cmk-1* mutant larvae grown on live bacteria and no exogenous pheromone (*Figure 2B,C*). The strongly reduced expression under fed conditions in *cmk-1* larvae precluded us from determining whether starvation or addition of pheromone resulted in a further decrease in *daf-7*p::*gfp* expression levels in this mutant background (*Figure 2B*). Expression of *cmk-1* in ASI, but not in AWC, rescued the *daf-7*p::*gfp* expression defects of *cmk-1* mutant larvae (*Figure 2C*), indicating that CMK-1 acts cell-autonomously in ASI to regulate *daf-7* expression.

We next asked whether CMK-1 acts in parallel to the TGF-β pathway to also regulate ILP gene expression. A *daf-28*p::*gfp* reporter is expressed strongly in both ASI and ASJ neurons in fed wild-type larvae (*Li et al., 2003*) (*Figure 3A,B*). Under our conditions, addition of crude pheromone decreased expression primarily in ASI, whereas starvation resulted in decreased expression in both ASI and ASJ (*Figure 3A*). Addition of pheromone to starved wild-type larvae did not appear to further decrease expression in either cell type (*Figure 3A*). Expression of *daf-28*p::*gfp* was unaffected by temperature (*Figure 3—figure supplement 1*). Interestingly, we observed that in fed *cmk-1* mutant larvae grown on live bacteria without exogenous pheromone, *daf-28*p::*gfp* expression was strongly decreased in ASJ and more weakly affected in ASI (*Figure 3A,B*). As in wild-type larvae, addition of pheromone or starvation decreased *daf-28*p::*gfp* expression in ASI; we were unable to detect whether expression in ASJ was further reduced under these conditions in *cmk-1* mutants (*Figure 3A*). The *daf-28*p::*gfp* expression defect of *cmk-1* mutants in ASJ was partly but significantly rescued upon expression of wild-type *cmk-1* sequences in AWC but not in ASI/AWA or ASJ (*Figure 3B*). These results indicate that while pheromone regulates *daf-28* expression primarily in ASI, food-dependent regulation of *daf-28* is observed in both ASI and ASJ. Moreover, we find that CMK-1 acts non cell-autonomously in AWC to regulate *daf-28* ILP gene expression specifically in ASJ under well-fed conditions.

If reduced expression of *daf-7* and *daf-28* in *cmk-1* mutants is causal to their dauer formation phenotypes, we would predict that increased expression of *daf-7* or *daf-28* would rescue the dauer formation defects of *cmk-1* mutants. We found that constitutive expression of *daf-7* and *daf-28* in ASI and ASJ, respectively, rescued the dauer formation phenotype of *cmk-1* mutants (*Figures 2D*, *3C*; *Figure 3—source data 1*). *daf-28* expression in ASI failed to rescue (*Figure 3C*) indicating that neuron-specific release properties or spatial diffusion constraints may require DAF-28 expression in ASJ to rescue the *cmk-1* phenotype (*Cornils et al., 2011*; *Chen et al., 2013*). Taken together, these results indicate that CMK-1 may relay food information into the dauer regulatory pathway by acting cell-autonomously to regulate *daf-7* TGF-β expression in ASI, and non cell-autonomously in AWC to regulate *daf-28* ILP expression in ASJ.

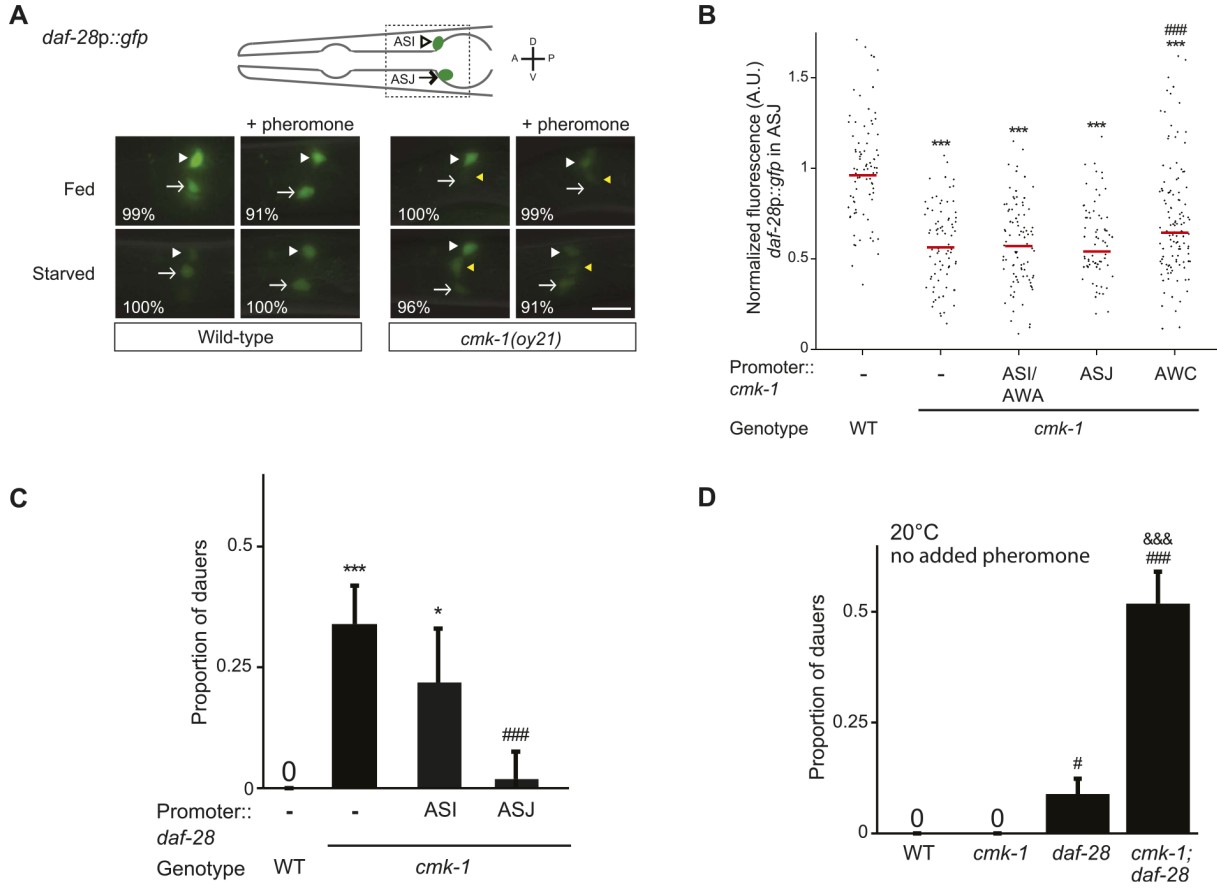

**Figure 3.** CMK-1 acts non cell-autonomously in AWC to regulate *daf-28* insulin-like peptide (ILP) gene expression in ASJ. (**A**) Representative images of *daf-28*p::*gfp* expression in L1 larvae of wild-type or *cmk-1(oy21)* animals under the indicated conditions. Schematic of worm head indicating positions of the ASI and ASJ sensory neuron cell bodies is shown at top; boxed region is shown in panels below. Animals were grown with plentiful live OP50 or starved for at least 6 hr in the absence or presence of 1 unit of crude pheromone (see 'Materials and methods'). White arrowheads and arrows indicate cell bodies of ASI and ASJ, respectively. Yellow arrowheads indicate expression in an ectopic cell observed in ~14% of wild-type and ~50% of *cmk-1* mutants under all conditions. Numbers in bottom left hand corners indicate the percentage of examined larvae that exhibit the shown expression patterns; n > 50 each; three independent experiments. Lateral view; scale bar: 10 μm. (**B**) Scatter plot of fluorescence intensity of *daf-28*p::*gfp* expression in ASJ in wild-type or *cmk-1(oy21)* mutants. Median is indicated by red horizontal line. Animals were grown on ample live OP50 in the absence of exogenous pheromone. Each dot is the fluorescence intensity of a single neuron in a given experiment; n > 60 neurons total each, at least three independent experiments. For transgenic strains, a representative line was selected from experiments shown in *Figure 1E* and crossed into the reporter strains. Promoters driving wild-type *cmk-1* cDNA were: ASI/AWA—*gpa-4*p; AWC—*ceh-36Δ*p; ASJ—*trx-1*p. (**C**) Dauers formed by shown strains on 80 μg live OP50 and 6 μM ascr#3. Promoters used to drive expression of wild-type *daf-28* cDNA were: ASI—*srg-47*p and ASJ—*trx-1*p. Shown are the averages of ≥3 independent experiments with >65 animals each. (**D**) Dauers formed by the indicated strains on 80 μg OP50 at 20°C in the absence of exogenous pheromone. Alleles used were: *cmk-1(oy21)* and *daf-28(tm2308)*. Shown are the averages of ≥3 independent experiments with >65 animals each. Errors are SEM. Unless indicated otherwise, * and *** indicate different from wild-type at p < 0.01 and 0.001, respectively; # and ### indicate different from *cmk-1* at p < 0.05 and 0.001, respectively; &&& indicates different from *daf-28* at p < 0.001 (ANOVA and Games-Howell post-hoc test).

The following source data and figure supplement are available for figure 3:

**Source data 1.** Dauer assay data for individual trials in *Figure 3*.
**Figure supplement 1.** *daf-28* expression is not affected by cultivation temperature.

We further tested the regulatory relationship between CMK-1 and the parallel TGF-β and DAF-28 ILP signaling pathways by performing genetic epistasis experiments. If CMK-1 acts in both the TGF-β and DAF-28 pathways, we predicted that *cmk-1; daf-28* double mutants would exhibit increased dauer formation defects compared to either single mutant alone, in part due to reduced *daf-7*

expression in *cmk-1* mutants. As shown in *Figure 3D*, we found that a significantly larger percentage of *cmk-1; daf-28* double mutant animals entered into the dauer stage as compared to *cmk-1* or *daf-28* null mutants alone in the presence of live food and no added pheromone. Taken together, these results confirm that CMK-1 regulates both TGF-β and insulin signaling to modulate dauer formation.

## Feeding state is reflected in the temporal dynamics of CMK-1 subcellular localization in AWC

The AWC neurons have not previously been implicated in dauer formation. We further explored the role of CMK-1 in AWC in the regulation of dauer formation as a function of feeding state. We and others previously showed that CMK-1 shuttles between the nucleus and the cytoplasm in thermosensory neurons based on growth temperature (*Schild et al., 2014*; *Yu et al., 2014*), and regulates the expression of thermotransduction genes (*Yu et al., 2014*). We asked whether CMK-1 subcellular localization in AWC is similarly affected by feeding status. In fed L1 larvae, a functional CMK-1::GFP fusion protein was present mostly uniformly throughout the soma of the AWC neurons (*Figure 4A, B*). Food withdrawal for 30 min resulted in a transient nuclear enrichment of CMK-1 that persisted for approximately 1 hr (*Figure 4A,B*). However, after prolonged starvation, CMK-1::GFP was enriched in the cytoplasm of AWC (*Figure 4A,B*). In contrast, food did not affect subcellular localization of CMK-1::GFP in AFD (*Figure 4A*), and we previously showed that temperature does not affect CMK-1 localization in AWC (*Yu et al., 2014*). Overexpression of a constitutively nuclear-enriched CMK-1::GFP protein in AWC strongly rescued the dauer formation phenotype of *cmk-1* mutants (*Figure 4C*; *Figure 4—source data 1*), whereas a constitutively cytoplasmically enriched protein rescued more weakly (*Figure 4C*). The subcellular localization of these proteins in AWC was unaffected by genotype or feeding state (*Figure 4—figure supplement 1*). However, we note that neither the nuclear-enriched nor the cytoplasmically enriched CMK-1::GFP fusion proteins are fully excluded from the cytoplasmic or nuclear compartment, respectively, possibly due to overexpression (*Figure 4—figure supplement 1*). These results suggest that feeding state regulates CMK-1 subcellular localization in AWC, and that CMK-1 activity in the nuclei of AWC may be important to encode food information in the dauer regulatory pathway.

## CMK-1 regulates the expression of ILP genes in AWC to antagonize dauer formation

We next investigated the nature of the CMK-1-regulated signal in AWC that may transmit food information to the dauer regulatory pathway. The transient nuclear localization of CMK-1 in AWC upon food withdrawal suggests that CMK-1 regulates gene expression in AWC as a function of feeding state. A link between nutrient availability and insulin signaling is now well established in many organisms (*Erion and Sehgal, 2013*; *Riera and Dillin, 2015*), and in *C. elegans*, the expression of several ILP genes has been shown to be regulated by food availability (*Li et al., 2003*; *Cornils et al., 2011*; *Ritter et al., 2013*; *Chen and Baugh, 2014*). Together with the hours-long timescale of integration of sensory cues for the regulation of dauer formation (*Golden and Riddle, 1984c*; *Schaedel et al., 2012*), and the fact that AWC may signal feeding state to regulate downstream hormonal signaling, we hypothesized that CMK-1 may regulate the expression of ILP genes in AWC as a function of feeding state.

Of the subset of 40 predicted ILP genes (*Pierce et al., 2001*; *Li and Kim, 2010*) (www.wormbase.org) whose expression is reported to be regulated by nutrient availability in *C. elegans*, only *ins-26* and *ins-35* are known to be expressed in AWC (as well as in additional cells) (*Chen and Baugh, 2014*). However, whether their expression in AWC is regulated by food has not been previously determined. We asked whether *ins-26* and *ins-35* expression in AWC is regulated by food and CMK-1. Under fed conditions, *ins-26*p::*yfp* was expressed in ASI, ASE, and a subset of AWC neurons in wild-type L1 larvae; expression in AWC, and to a lesser extent in ASI, was reduced upon starvation (*Figure 5A,B*). Relative to wild-type, *ins-26*p::*yfp* expression was decreased in ASE and AWC in *cmk-1* mutants grown with plentiful food, whereas expression in ASI decreased only upon starvation (*Figure 5A,B*). Similarly, *ins-35*p::*yfp* was largely expressed in ASE and AWC in fed wild-type L1 larvae; expression in both neurons was significantly decreased upon starvation, and in fed and starved *cmk-1* mutants (*Figure 5A,B*). Weak expression of *ins-35* was also observed in the ASK neurons, particularly in *cmk-1* mutants, although expression in this neuron type was not further modulated by

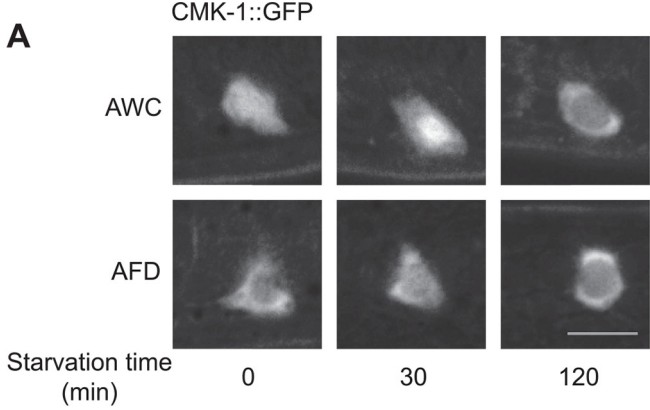

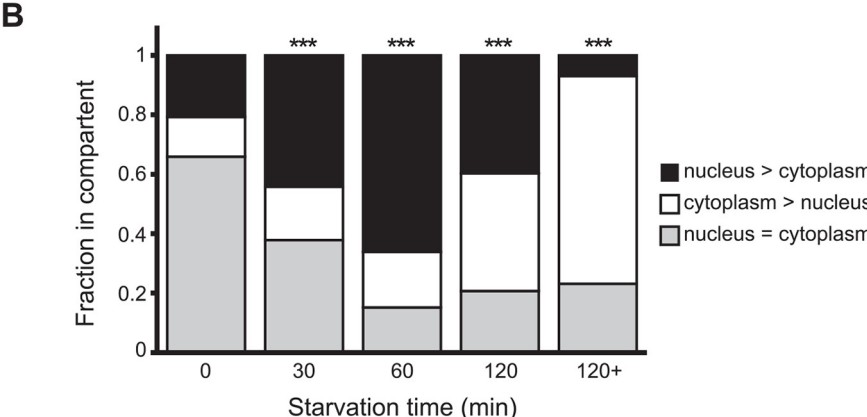

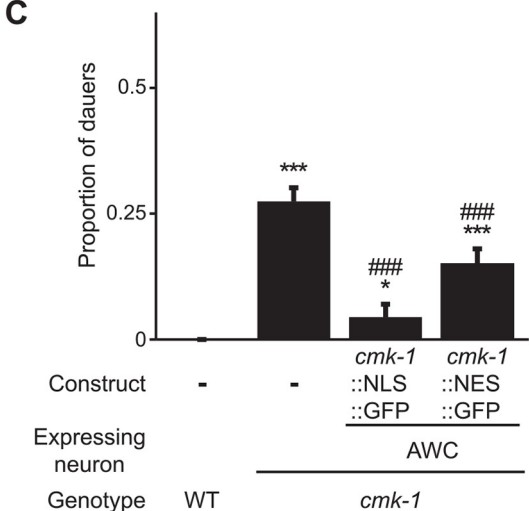

**Figure 4.** Subcellular localization of CMK-1 in AWC is regulated by feeding state. (**A**, **B**) Representative images (**A**) and quantification (**B**) of subcellular localization of CMK-1::GFP in AWC neurons following removal from food for the indicated times. Representative images of CMK-1::GFP localization in AFD are also shown in **A**. Scale bar: 5 μm (**A**). n > 75 AWC neurons each (**B**). *** indicates different from distribution at 0 min at $p < 0.001$ ($\chi^2$ test). (**C**) Dauers formed by the indicated strains on 80 μg live OP50 and 6 μM ascr#3. CMK-1::NLS::GFP and CMK-1::NES::GFP were expressed in AWC under the *ceh-36Δ* promoter. Shown are the averages of ≥3 independent experiments with >65 animals each. For transgenic strains, data are averaged from two independent lines each. Errors are SEM. *and *** indicate different from wild-type at $p < 0.05$ and 0.001, respectively; ### indicates different from *cmk-1(oy21)* at $p < 0.001$ (ANOVA and Games-Howell post-hoc test).

The following source data and figure supplement are available for figure 4:

**Source data 1.** Dauer assay data for individual trials in *Figure 4*.

*Figure 4 continued on next page*

*Figure 4 continued*

**Figure supplement 1.** Localization of CMK-1::NLS::GFP and CMK-1::NES::GFP in AWC.

starvation (*Figure 5B*). Expression of a constitutively nuclear-enriched CMK-1 protein in AWC neurons increased *ins-35* expression in *cmk-1* mutants in both fed and starved conditions, to a slightly greater extent than a constitutively cytoplasmically enriched protein (*Figure 5B*). These observations suggest that the expression of *ins-26* and *ins-35* in AWC is regulated by food availability, and that their expression pattern in AWC in fed *cmk-1* mutants resembles that of starved wild-type animals. Moreover, nucleocytoplasmic shuttling of CMK-1 may be important for feeding state-dependent regulation of *ins-35* gene expression.

Reduced expression of *ins-26* and *ins-35* in AWC in starved wild-type or fed *cmk-1* mutants suggest that these peptides are candidates for an anti-dauer or pro-growth signal from AWC. If this were the case, we would predict that loss of function of these genes would enhance dauer formation in wild-type but not in *cmk-1* mutants. Indeed, we found that *ins-26; ins-35* double mutants formed more dauers on heat-killed bacteria in the presence of pheromone than wild-type animals (*Figure 5C*; *Figure 5—source data 1*). Dauer formation in *cmk-1* mutants on live food was not significantly enhanced upon loss of either *ins-26* or *ins-35* (*Figure 5D*). Since live vs heat-killed food provide different sensory inputs that might confound comparisons between wild-type and *cmk-1* mutants, we established conditions under which we could force a small fraction of wild-type larvae to enter into the dauer stage on live food (see 'Materials and methods'). *ins-26; ins-35* double mutants also exhibited enhanced dauer formation on live food (*Figure 5C*), further suggesting that these ILPs may comprise an anti-dauer signal.

To confirm that reduced expression of these ILP genes in AWC in *cmk-1* mutants is partly causal to their increased dauer formation phenotype, we asked whether restoration of ILP gene expression specifically in AWC was sufficient to rescue the dauer formation phenotype of *cmk-1* mutants. As shown in *Figure 5D*, we found that overexpression of *ins-26* and *ins-35* specifically in AWC, but not in ASE or ASJ, rescued the dauer formation defect of *cmk-1* mutants. Overexpression of *ins-26* and *ins-35* in AWC in *cmk-1* mutants was also sufficient to partly restore *daf-28p::gfp* expression in ASJ (*Figure 5E*). Together, these results suggest that food-dependent regulation of *ins-26* and *ins-35* in AWC may comprise an anti-dauer signal, and that inappropriate downregulation of these genes in AWC in *cmk-1* mutants may in part cause the increased dauer formation phenotype in these animals under fed conditions.

## The balance between anti- and pro-dauer signals from AWC is disrupted in *cmk-1* mutants

If AWC were in part required to transmit information about food availability to limit dauer formation, we would predict that ablation of AWC would result in increased dauer formation in wild-type animals, and that dauer formation in these animals would be less sensitive to regulation by food. Indeed, we found that AWC-ablated animals formed more dauers overall on heat-killed food than wild-type animals, and moreover, the ability of heat-killed bacteria to modulate dauer formation was reduced (*Figure 5F*).

Since the *ins-26* and *ins-35* anti-dauer signals are downregulated in *cmk-1* mutants, we expected that ablation of AWC would not further increase dauer formation in *cmk-1* mutants. Unexpectedly, we found that ablation of AWC instead suppressed dauer formation in *cmk-1* mutants grown on live food (*Figure 5G*). One interpretation of this observation is that AWC also sends a pro-dauer signal in *cmk-1* mutants. Consistent with this notion, ablation of AWC significantly rescued *daf-28p::gfp* expression in ASJ in *cmk-1(oy21)* animals (*Figure 5—figure supplement 1*). We considered the possibility that this pro-dauer signal could be present aberrantly only in *cmk-1* mutants. Alternatively, wild-type AWC neurons could also send a pro-dauer signal, but this signal may be masked by the different food conditions (heat-killed vs live food) employed to examine dauer formation in wild-type and *cmk-1* mutants, respectively. To address this issue, we asked whether ablation of AWC alters dauer formation of wild-type animals on live food. Indeed, we found that ablation of AWC also

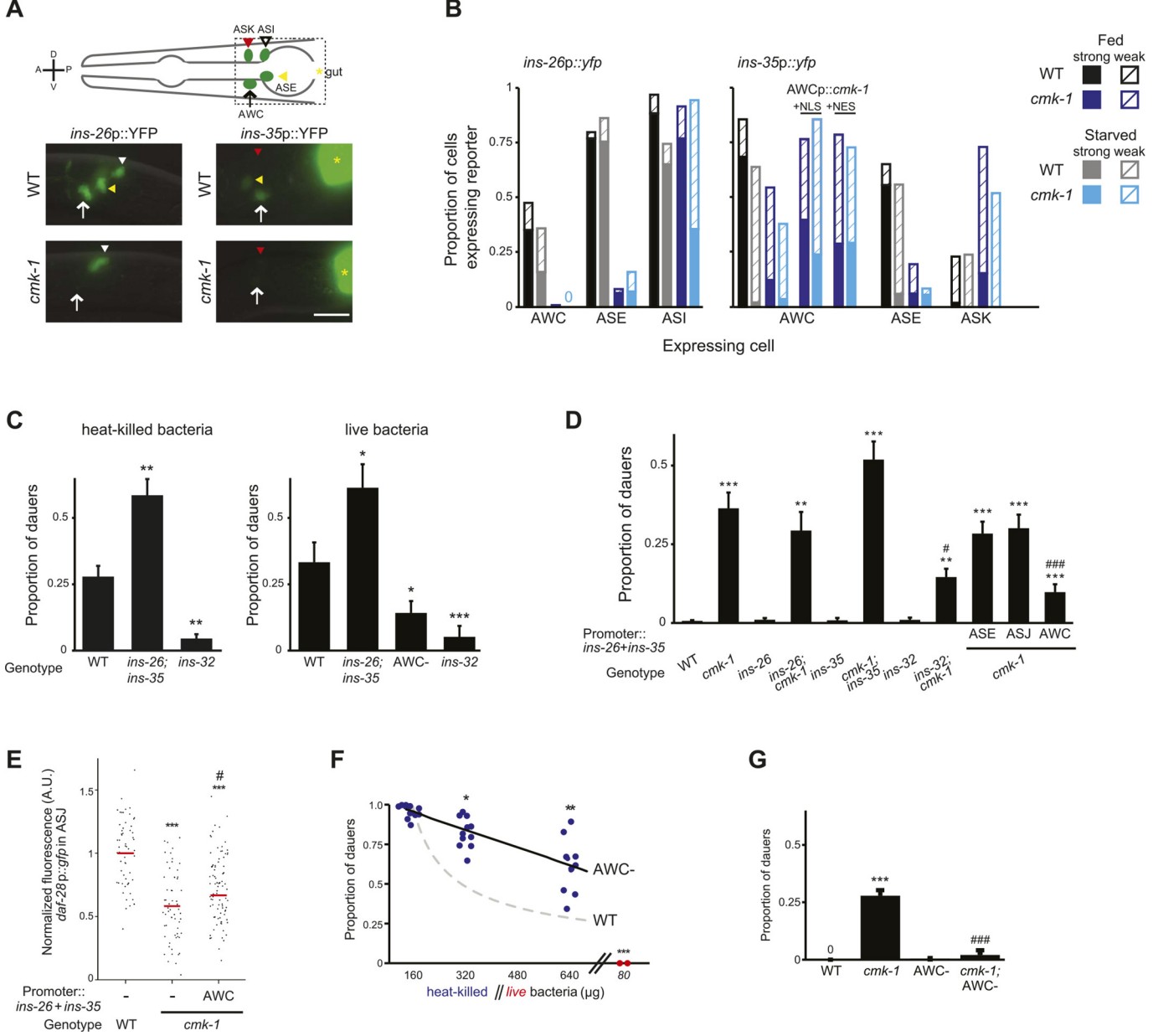

**Figure 5.** CMK-1 maintains a balance of anti- and pro-dauer signals from AWC as a function of feeding state. (**A**) (Top) Schematic of worm head indicating positions of sensory neuron soma. Boxed area is shown in images below. (Below) Representative images of *ins-26*p::*yfp* and *ins-35*p::*yfp* expression in fed wild-type and *cmk-1(oy21)* mutants. White and yellow arrowheads indicate ASI and ASE, respectively; white arrow indicates AWC; yellow asterisk marks expression in the intestine. The location of ASK is indicated by a red arrowhead; fluorescence in ASK is weak and not visible at this exposure in shown images. Lateral view; scale bar: 10 μm. (**B**) Quantification of expression in each neuron type in fed and starved (>6 hr) conditions. Solid and hatched bars indicate strong and weak expression, respectively, in each cell. n > 50 animals each; three independent experiments. (**C**) Dauers formed by shown strains on 160 μg heat-killed OP50 and 60 nM ascr#2 (left), and 80 μg live OP50 and 6 μM ascr#3 + 600 nM ascr#5 (right). Alleles used were: *ins-26(tm1983)*, *ins-35(ok3297)* and *ins-32(tm6109)*. n > 4 assays of 65 animals each; at least three independent experiments. (**D**) Dauers formed by shown strains on 6 μM ascr#3 and 80 μg live OP50. *ins-26* and *ins-35* cDNAs were expressed in ASE, ASJ, and AWC under *che-1*, *trx-1*, and *ceh-36*Δ regulatory sequences, respectively. At least two independent lines were analyzed for each transgenic strain. Shown are the averages of at least three independent experiments with >65 animals each. (**E**) Scatter plot of fluorescence intensity of *daf-28*p::*gfp* expression in ASJ in the indicated genetic backgrounds. *ins-26* and *ins-35*::SL2::*mCherry* cDNAs were expressed in AWC under the *odr-1* promoter. Only animals expressing mCherry in AWC were scored. Median is indicated by red horizontal line. Each dot is the fluorescence intensity of a single neuron in a given experiment. n > 60 neurons total each, four independent experiments. (**F**) Quantification of dauer formation in AWC-ablated animals in the presence of 6 μM ascr#3 and the indicated amounts of heat-killed (blue circles) or live (red circles) OP50. Each filled circle represents one assay; n > 65 animals per assay, five independent experiments. Line represents best fit to the data. Dashed line indicates the curve for wild-type animals from *Figure 1B* shown for comparison. *, **, and *** indicate different from values using 160 μg of heat-killed bacteria at p < 0.05, 0.01, and 0.001, respectively. (**G**) Dauers

*Figure 5 continued on next page*

*Figure 5 continued*

formed by the indicated strains on 80 μg live OP50 and 6 μM ascr#3. For each assay: n > 65 animals; four independent experiments. Errors are SEM. Except where indicated, *, **, and *** indicate different from wild-type at p < 0.05, 0.01, and 0.001, respectively; #, ##, and ### indicate different from *cmk-1(oy21)* at p < 0.05, 0.01, and 0.001, respectively (ANOVA and Games-Howell post-hoc test).

The following source data and figure supplements are available for figure 5:

**Source data 1.** Dauer assay data for individual trials in *Figure 5*.

**Source data 2.** Dauer assay data for individua reliably quantify the effects of lower concentrations of live bactl trials in *Figure 5—figure supplement 2*.

**Figure supplement 1.** Scatter plot of fluorescence intensity of *daf-28p::gfp* expression in ASJ.

**Figure supplement 2.** CMK-1 regulates a BLI-4-dependent pro-dauer signal from AWC.

decreased dauer formation in wild-type animals grown on live food (*Figure 5C*), and increased *daf-28*p::*gfp* expression in ASJ (*Figure 5—figure supplement 1*) suggesting that AWC also sends a pro-dauer signal in wild-type animals that is revealed under specific conditions.

We next investigated the identity of this pro-dauer signal. The BLI-4 and EGL-3 proprotein convertases regulate the processing of different subsets of ILP precursors (*Leinwand and Chalasani, 2013*; *Hung et al., 2014*). Knocking down *bli-4*, but not *egl-3* in the AWC neurons suppressed dauer formation in *cmk-1* mutants (*Figure 5—figure supplement 2*; *Figure 5—source data 2*), suggesting that a BLI-4-dependent insulin signal(s) from AWC promotes dauer formation in *cmk-1* mutants. Of the nineteen predicted BLI-4 targets (*Leinwand and Chalasani, 2013*), only the *ins-32* ILP gene was previously reported to be expressed in AWC in adult hermaphrodites (*Takayama et al., 2010*). We found that *ins-32* mutants formed fewer dauers on heat-killed food, as well as live food, and pheromone in an otherwise wild-type background (*Figure 5C*). Loss of *ins-32* also suppressed the increased dauer formation phenotype of *cmk-1* mutants on live food and pheromone (*Figure 5D*). We could not detect *ins-32* expression in AWC in L1 larvae although we noted that expression of *ins-32* reporter gene was weak and dynamic (SJN and PS, unpublished), raising the possibility that *ins-32* is expressed in AWC only transiently during the period of dauer signal integration in early postembryonic development (*Golden and Riddle, 1984c*; *Schaedel et al., 2012*). Together, these observations suggest that different insulin peptides comprise the food-dependent anti- and pro-dauer signals from AWC, and that the balance between expression and/or release of these antagonistic signals is disrupted in *cmk-1* mutants.

## The AWC neurons are hyperactive upon long-term starvation in wild-type animals and in fed *cmk-1* mutants

CaMKs such as CaMKI and CaMKIV play critical roles in regulating neuronal activity-dependent processes, including activity-dependent gene expression (*Wayman et al., 2008*, *2011*; *Cohen et al., 2015*). In turn, changes in gene expression feed back to regulate neuronal state and properties. We asked whether AWC activity is altered upon starvation or in *cmk-1* mutants. It has previously been shown that the AWC neurons in adult animals are silenced in the presence of food or food-related odorants and are activated upon food/odor withdrawal (*Chalasani et al., 2007*). Consistent with this observation, the AWC neurons of well-fed wild-type L2 larvae expressing the GCaMP calcium indicator (*Nakai et al., 2001*) exhibited few, if any somatic calcium transients (*Figure 6A,B*). However, we found that wild-type AWC neurons became hyperactive following prolonged food deprivation. Both the frequency and amplitude of spontaneous calcium transients increased upon starvation for 2 hr, although no effects were observed after 1 hr (*Figure 6A,B*). Interestingly, the AWC neurons in well-fed *cmk-1* mutant larvae exhibited a high frequency and amplitude of somatic calcium transients similar to the activity pattern observed in starved wild-type larvae (*Figure 6A,B*); activity in AWC in *cmk-1* larvae was not further increased by starvation (*Figure 6A,B*). Overexpression of wild-type *cmk-1* sequences in AWC suppressed basal GCaMP expression (data not shown), implying that

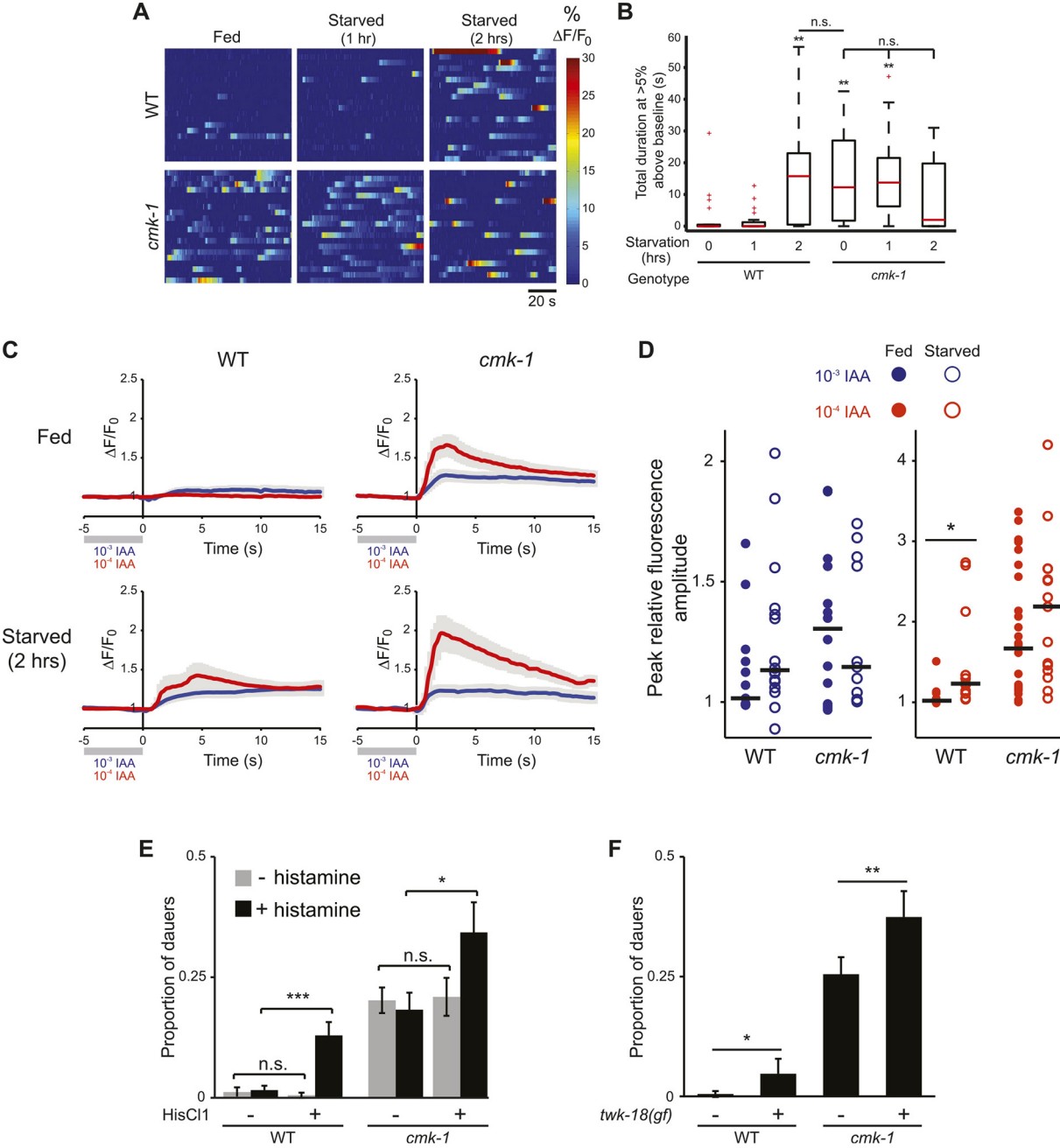

**Figure 6.** The AWC neurons exhibit increased basal activity in fed *cmk-1*, and starved wild-type animals. (**A**) Heat maps showing the fluorescence intensity ($\Delta F/F_0$) in the soma of AWC neurons in fed or starved wild-type and *cmk-1(oy21)* L2 larvae expressing GCaMP 3.0 in AWC under the *ceh-36Δ* promoter. Animals were cultured at 20°C, starved for 0, 1, or 2 hr and imaged at 20°C. Each horizontal line shows calcium dynamics in a single AWC neuron; n = 20 neurons each. (**B**) Box-and-whisker plots quantifying total duration of calcium responses >5% above baseline for each genotype and condition shown in **A**. Median is indicated by a red line. Tops and bottoms of boxes indicate the 75th and 25th percentiles, respectively; whiskers represent fifth and 95th percentiles. Outliers are indicated by + signs. ** indicates different from wild-type at 0 hr at p < 0.01 (Kruskal–Wallis test). n.s.—not significant. (**C**) Average calcium responses in AWC neurons of fed and starved animals expressing GCaMP 3.0 under the *ceh-36Δ* promoter in the presence, or upon removal of the odorant isoamyl alcohol (IAA), diluted to $10^{-3}$ (blue) or $10^{-4}$ (red). Error bars are SEM and are represented by light gray shading. n ≥ 10 for each genotype and condition shown. (**D**) Scatter plot of the peak fluorescence amplitudes of individual neuron responses following odorant removal for the indicated conditions and genotypes. Horizontal black bars represent the median. * represents different between the indicated values at p < 0.05 (Student's *t*-test). (**E**) Dauer formation in the presence of 80 µg live OP50 and 6 µM ascr#3 by non-transgenic and transgenic animals expressing the *Drosophila* histidine-gated chloride channel 1 (HisCl1) in the presence (black) or absence (gray) of 10 mM histidine. Error bars are SEM. n.s.—not significant; * and *** indicate different between indicated values at p < 0.05 and 0.001, respectively (Student's *t*-test). (**F**) Dauer

*Figure 6 continued on next page*

*Figure 6 continued*

formation in the presence of 80 μg live OP50 and 6 μM ascr#3 by non-transgenic and transgenic animals expressing the constitutively active potassium channel TWK-18(*gf*). Error bars are SEM. * and ** indicate different between indicated values at p < 0.05 and 0.01, respectively (Student's *t*-test).

The following source data and figure supplement are available for figure 6:

**Source data 1.** Dauer assay data for individual trials in *Figure 6*.

**Figure supplement 1.** AWC neurons exhibit increased responses to odorant removal in fed *cmk-1* and starved wild-type animals.

AWC may be silenced under these conditions, and precluded examination of cell-specific effects of CMK-1 on AWC activity.

Fed wild-type animals must be transiently starved for the AWC neurons to respond to odor/food removal following a brief exposure to the stimulus (S Chalasani, personal communication). Confirming this observation, we found that removal of either of the attractive odorants isoamyl alcohol or benzaldehyde following a 1 min exposure failed to elicit a response in the AWC neurons of wild-type adult animals transferred directly from food, but led to responses in animals starved on a food-free plate (*Figure 6C,D*, *Figure 6—figure supplement 1*). Since the activity state of AWC in fed *cmk-1* mutants resembles that of starved wild-type animals, we asked whether AWC neurons in *cmk-1* mutant adults are able to respond to odor removal without prior starvation. Indeed, the AWC neurons in *cmk-1* adult animals responded robustly to odorant removal regardless of their feeding state (*Figure 6C,D*, *Figure 6—figure supplement 1*). Together, these observations imply that the basal activity state of the AWC neurons in fed *cmk-1* mutants is similar to the activity state of these neurons in wild-type animals starved for extended periods. Importantly, these results also indicate that AWC retains the ability to respond to food-associated odors in *cmk-1* mutants.

We next explored the relationship between increased AWC activity in *cmk-1* mutants and their enhanced dauer phenotype. To address this issue, we inhibited activity in AWC neurons and examined the effect of this inhibition on dauer formation. Addition of exogenous histamine has been shown to silence neuronal activity in *C. elegans* neurons expressing the *Drosophila* histamine-gated chloride (HisCl1) channel (*Pokala et al., 2014*). Expression of an activated potassium channel (*twk-18 (gf)*) has also been shown to hyperpolarize *C. elegans* neurons (*Kunkel et al., 2000*; *Kawano et al., 2011*; *Zhang and Zhang, 2012*). We found that growth on 10 mM histamine increased dauer formation in transgenic wild-type as well as in *cmk-1* mutants expressing HisCl1 in AWC but had no effect on non-transgenic animals (*Figure 6E*; *Figure 6—source data 1*). Similarly, dauer formation was enhanced in both *cmk-1(oy21),* and to a weaker extent in wild-type animals expressing *twk-18(gf)* in AWC (*Figure 6F*). These results suggests that activity in AWC antagonizes dauer formation in both wild-type and *cmk-1* mutants, and acts in parallel with CMK-1 to regulate dauer formation (*Figure 7*; see 'Discussion').

## Discussion

### A balance between anti- and pro-dauer ILP signals from AWC relays food inputs into the dauer decision

Environmental signals such as food availability are encoded in the level of expression of the *daf-7* TGF-β, and *daf-28* and other ILP genes, to regulate the binary decision between entry into the reproductive cycle or the dauer stage (*Ren et al., 1996*; *Schackwitz et al., 1996*; *Li et al., 2003*; *Entchev et al., 2015*). We have identified AWC as key neurons that translate food information into changes in *daf-28* ILP gene expression to regulate dauer formation. Our results suggest a model in which the AWC neurons transmit both anti- and pro-dauer signals; the balance between these signals is determined by food inputs and feeding state to regulate a critical developmental decision in the nematode life cycle. We show that the anti-dauer signals are comprised in part of the AWC-expressed *ins-26* and *ins-35* ILP genes, whereas the *ins-32* ILP gene may constitute a component of the pro-dauer signal (*Figure 7*). Expression of *ins-26* and *ins-35* in AWC is downregulated under

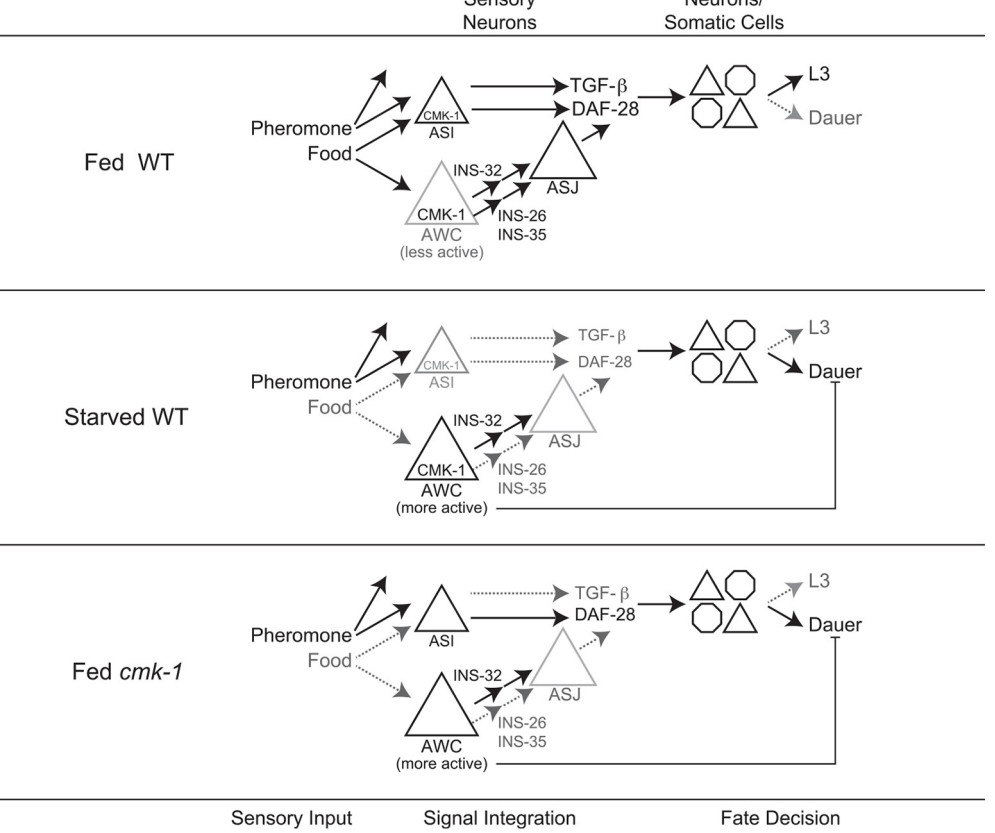

**Figure 7.** Model for the role of AWC and CMK-1 in the regulation of the dauer decision as a function of feeding state. CMK-1 acts in AWC to drive expression of the anti-dauer *ins-26* and *ins-35* ILP genes. CMK-1 may also regulate expression of the *ins-32* (or other) pro-dauer signal in AWC; alternatively, the pro-dauer signal could be present tonically regardless of environmental state. Under fed conditions, CMK-1-regulated anti-dauer signals from AWC predominate and drive expression of the *daf-28* ILP gene in ASJ. CMK-1 also acts cell-autonomously to regulate expression of *daf-7* TGF-β in ASI. Together, *daf-7* TGF-β and *daf-28* ILP signals promote reproductive development. When starved, anti-dauer signals from AWC are downregulated resulting in decreased expression of *daf-28* in ASJ. *daf-7* and *daf-28* expression in ASI are also downregulated upon starvation. The shifted balance towards pro-dauer signals promotes dauer formation in the presence of pheromone. In *cmk-1* mutants, loss of the anti-dauer signals from AWC downregulates *daf-28* expression in ASJ under fed conditions. *daf-7* expression in ASI is also downregulated in *cmk-1* mutants. Consequently, the inappropriate predominance of pro-dauer signals in *cmk-1* mutants promotes dauer formation in the presence of plentiful food and pheromone. Feeding conditions also modulate AWC basal neuronal activity; increased activity upon starvation or in *cmk-1* mutants may limit dauer formation via a parallel pathway.

starvation conditions that promote dauer formation, and loss of both genes enhances dauer formation (*Figure 7*). Consistent with a role for these ILPs in antagonizing dauer formation, *ins-26* and *ins-35* mutations were reported to enhance dauer formation in sensitized backgrounds, and *ins-35* was proposed to represent a genetic 'hub' regulating dauer entry (*Fernandes de Abreu et al., 2014*). Conversely, loss of *ins-32* suppresses dauer formation, although it remains possible that *ins-32* acts in cell types other than AWC to promote dauer formation.

The complex balance between pro- and anti-dauer signals as a function of environmental conditions is highlighted in the effects of AWC ablation on dauer formation. Ablation of AWC enhances dauer formation on heat-killed food but suppresses dauer formation on live food. Bacterial food is a complex cue consisting of many volatile and non-volatile compounds (*Orth et al., 2011*) that are sensed by multiple neuron types. Live and heat-killed bacteria provide different signals which may be sensed and integrated by distinct networks in the dauer decision. Thus, integration of different food signals from diverse sensory neurons likely differentially influences dauer formation on different food types. Indeed, ASI detects food directly to regulate *daf-7* expression (*Ren et al., 1996*; *Schackwitz et al., 1996*; *Gallagher et al., 2013*), and recent work has shown that complex patterns

and shapes of expression of *daf-7* and the *tph-1* tryptophan hydroxylase genes across multiple neuron types represents a 'neural code' for food abundance in the context of regulation of adult lifespan (*Entchev et al., 2015*). Given the consequences of phenotypic plasticity on fitness (*Nijhout, 2003*; *Avery, 2014*), the precise tuning of anti- and pro-dauer signals in multiple neurons, including the AWC neurons, is critical to allow animals to make the optimal developmental decision in response to specific environmental conditions.

## CMK-1 regulates the balance between anti- and pro-dauer signals in AWC as a function of feeding state

We have shown that the CMK-1 CaMKI plays a critical role in regulating the balance between anti- and pro-dauer signals from AWC. Our results suggest that in *cmk-1* mutants, the balance between these signals is decoupled from feeding state, resulting in deregulated dauer formation under conditions that suppress dauer formation in wild-type animals. This hypothesis is based on several lines of evidence. First, *cmk-1* mutants form dauers inappropriately on food conditions that fully suppress dauer formation in wild-type animals. Second, the expression patterns of *daf-7* in ASI and *daf-28* in ASJ in fed *cmk-1* mutants are similar to those in starved wild-type animals, indicating that the expression of these key dauer-regulatory hormones does not reflect food abundance accurately in the absence of CMK-1. Third, expression of *ins-26* and *ins-35* ILP genes in AWC in fed *cmk-1* mutants resembles those in starved wild-type animals, further supporting the notion that food signals and gene expression patterns are uncoupled in *cmk-1* mutants (*Figure 7*). Overexpression of *ins-26* and *ins-35* from AWC, and loss of *ins-32* suppress the enhanced dauer formation phenotype of *cmk-1* mutants indicating that the altered balance between anti-dauer and pro-dauer signals together with decreased *daf-7* expression in ASI, sensitizes the threshold for dauer formation in *cmk-1* mutants. Consistent with the notion that this balance is disproportionately shifted towards increased pro-dauer signals from AWC in *cmk-1* mutants, ablation of AWC suppresses increased dauer formation and partially restores *daf-28p::gfp* expression in this background. It is important to note, however, that the expression pattern changes in fed *cmk-1* mutants are not identical to those in starved wild-type animals, indicating that additional mechanisms translate food abundance into changes in gene expression patterns.

How might CMK-1 link food stimuli to dauer signals from AWC? We report that the subcellular localization of CMK-1 in AWC is highly dynamic and feeding state-dependent. Starvation results in transient nuclear localization of CMK-1, followed by cytoplasmic enrichment upon prolonged food deprivation. Although we do not yet know the nature of the signal that triggers the initial nuclear translocation, neuronal activity has been shown in multiple contexts to regulate subcellular localization of both mammalian and *C. elegans* CaMKI (*Eto et al., 1999*; *Ueda et al., 1999*; *Sakagami et al., 2005*; *Schild et al., 2014*; *Yu et al., 2014*). Since the AWC neurons must be transiently starved to exhibit responses to odorant removal, one possibility is that this odor removal-induced depolarization promotes CMK-1 nuclear translocation.

Consistent with a well-described role of CaMKs in mediating activity-dependent gene expression (*Flavell and Greenberg, 2008*; *Wayman et al., 2011*; *West and Greenberg, 2011*; *Bengtson and Bading, 2012*), nuclear localized CMK-1 may then regulate the transcription of anti- and pro-dauer signals. We show that constitutive nuclear localization of CMK-1 is sufficient to suppress dauer formation in the *cmk-1* background and increase *ins-35* expression under both fed and starved conditions. An intriguing possibility is that CMK-1 initially upregulates anti-dauer signals such as *ins-26* and *ins-35* upon transient food removal. As food deprivation persists, CMK-1 moves into the cytoplasm resulting in downregulation of anti-dauer signals to promote dauer formation. CMK-1 may also upregulate pro-dauer signals from AWC. Alternatively, pro-dauer signals may be expressed and/or released tonically, and CMK-1-mediated regulation of anti-dauer signals may be sufficient to promote or inhibit dauer formation. *C. elegans* larvae integrate food signals over an extended period of time to allow accurate assessment of their past, current, and future environment (*Schaedel et al., 2012*; *Avery, 2014*). We speculate that not only food abundance but also temporal variability in food availability must be encoded in the balance of anti- and pro-dauer signals. Such mechanisms would allow larvae to ignore relatively transient fluctuations in environmental cues and ensure that irreversible commitment to the dauer stage occurs only under persistently adverse conditions.

## Neuronal activity acts in parallel with CMK-1 to antagonize dauer formation

Basal activity in AWC is increased upon prolonged starvation in wild-type animals, and in fed *cmk-1* mutants. In wild-type animals, increased AWC activity upon prolonged starvation may be a consequence of feedback from internal state. Similar modulation of peripheral sensory neuron responses as a function of feeding state has been suggested to underlie state-dependent plasticity in sensory behaviors in other organisms (*Jyotaki et al., 2010*; *Palouzier-Paulignan et al., 2012*; *Sengupta, 2013*; *Pool and Scott, 2014*). In starved wild-type animals, or in the absence of CMK-1 function, this state-dependent feedback may increase AWC activity, and this increased activity in turn, may antagonize dauer formation by regulating an as yet uncharacterized pathway that acts in parallel to the CMK-1 regulated pathway in AWC. Why would animals need to continue to antagonize dauer formation even following prolonged starvation? Pre-dauer L2d larvae integrate environmental cues from 16 hr–33 hr post hatching and make an irreversible commitment to dauer entry only after 33 hr (*Schaedel et al., 2012*). Given this long timeframe of sensory signal integration, and that entry into the dauer stage is a bet-hedging strategy that maximizes fitness under uncertain environmental conditions (*DeWitt and Scheiner, 2004*; *Avery, 2014*), the availability of multiple dauer-regulatory pathways that integrate environmental cues on different timescales may be critical to correctly promote or suppress the dauer decision. A goal for the future will be to define how activity in AWC modulates dauer formation, to identify the circuit mechanisms by which AWC activity is altered as a function of starvation, and to correlate temporal changes in AWC activity with commitment to the dauer stage.

## Concluding remarks

Although dauer formation and other forms of polyphenism are the extreme examples of phenotypic plasticity, environmental cues experienced during defined sensitive or critical periods during development also underlie phenotypic variation in mammals (*Gluckman et al., 2007*). Our results describe how *C. elegans* integrates and translates environmental cues into hormonal signaling to regulate the dauer decision. We expect that continued investigation of the molecular and neuronal regulation of phenotypic plasticity by sensory cues in different species will lead to insights into the general principles underlying these critical developmental decisions, as well as provide information about the mechanisms of sensory integration that direct the choice of the appropriate developmental pathway.

# Materials and methods

## Strains and genetics

*C. elegans* was maintained on nematode growth medium (NGM) agar plates at 20°C, with *Escherichia coli* OP50 as a food source. Strains were constructed using standard genetic procedures. The presence of mutations was confirmed by PCR-based amplification and/or sequencing. A complete list of strains used in this study is shown in *Supplementary file 1*.

## Molecular biology

Promoter sequences and cDNAs were amplified from genomic DNA or a cDNA library, respectively, from a population of mixed-stage wild-type animals. cDNA sequences were verified by sequencing. The promoters used in this study are as follows: *cmk-1*p (3.1 kb), *sra-9*p (ASK, 2.9 kb), *ttx-1*p (AFD, 2.7 kb), *trx-1*p (ASJ, 1.0 kb), *gpa-4*p (ASI/AWA, 2.8 kb), *ceh-36Δ*p (AWC, 0.6 kb), *srg-47*p (ASI, 1.0 kb), *ins-26*p (see *Chen and Baugh, 2014*), *ins-35*p (see *Chen and Baugh, 2014*), *che-1*p (ASE, 0.7 kb), *odr-1*p (AWC [strong], AWB [weak], 2.4 kb) and *odr-3*p (AWC [strong], AWA, AWB, ASH, ADF [weak], 2.7 kb). Sense and antisense constructs for RNAi were generated by amplifying exons 2–8 (1.5 kb; *bli-4*) and 6–8 (568 bp; *egl-3*) from cDNAs, and cloning into vectors containing cell-specific promoter sequences. Linearized vectors containing either sense or antisense sequences were injected at 100 ng/μl each. The *twk-18(gf)* allele (*Kunkel et al., 2000*; *Kawano et al., 2011*; *Zhang and Zhang, 2012*) and the *Drosophila* HisCl1 channel cDNA (*Pokala et al., 2014*) fused via SL2 to an mCherry reporter were expressed under *ceh-36Δ*p regulatory sequences. *ceh-36Δ*p::HisCl1 and *ceh-36Δ*p::*twk-18(gf)* were injected at a concentration of 50 ng/μl.

## Dauer assays

Dauer assays were performed essentially as described (*Neal et al., 2013*), using the indicated food sources. Briefly, young adult worms were allowed to lay 65–85 eggs on an assay plate (*Neal et al., 2013*) containing either ethanol (control) or pheromone, and a defined amount of bacteria. Animals were grown at 25°C unless indicated otherwise. Assays using heat-killed food or live food were examined for dauer and non-dauer larvae approximately 84 hr or 66 hr, respectively, after the mid-point of the egg lay. In order to induce wild-type animals to form dauers on assay plates containing 80 µg live food, a mixture of 6 µM ascr#3 + 600 nM ascr#5 was used. For experiments involving transgenic animals expressing the *Drosophila* HisCl1 channel, 30 µl of 1 M histidine (Sigma-Aldrich, St. Louis, MO) was mixed with either pheromone or ethanol and was added to the assay plates and overlaid with molten assay agar such that the final concentration was 10 mM. At least three independent trials were conducted for each condition with two technical replicates each. Statistical analyses of dauer data were performed in SPSS (IBM, Armonk, NY), as described in the figure legends.

## Quantification of fluorescence intensity

Strains expressing fluorescent reporters were growth-synchronized by hypochlorite treatment, and embryos were allowed to develop for 20–24 hr at 20°C to the end of the L1 larval stage on OP50, in the presence or absence of crude pheromone (*Golden and Riddle, 1982*; *Zhang et al., 2013*). Crude pheromone plates were prepared by spreading 20 µl of 1:4 crude pheromone (~1 unit, defined as the amount required for forming ~33% dauers on heat-killed OP50 at 25°C) on the agar surface and allowing it to dry completely prior to seeding with bacteria. For starvation conditions, worms were washed from growth plates in M9 buffer and were transferred to assay plates for 4–6 hr. Prior to imaging, worms were collected by centrifugation, transferred to a 2% agarose pad on a microscope slide, and immobilized using 10 mM levamisole (Sigma-Aldrich). Animals were visualized on a Zeiss Axio Imager.M2 microscope using either a 40× (NA 1.3) or 63× (NA 1.4) oil objective, and images were captured using a Hamamatsu Orca camera. Neurons were identified by position relative to the subset of neurons filled with DiI (Sigma-Aldrich).

For quantification of fluorescence intensity, images were acquired from a single focal plane. The exposure time for each fluorescent reporter was adjusted in the wild-type background to ensure that pixel intensity in the cell of interest was in the linear range. Pixel intensities for the soma (*daf-7*p::*gfp* reporter) or the nucleus (*daf-28*p::*gfp* reporter) were measured in ImageJ (NIH) by calculating the mean pixel intensity for the entire region of interest. All measurements within a single experiment were normalized to the median wild-type expression value (set at 1) to account for variation in conditions across trials. All imaging and subsequent quantification was performed blind to the genotype.

For shown representative images, *z*-stacks (0.5 µm per slice) were acquired through the head of the animal, and a sub-stack containing all GFP-expressing cells was rotated and maximally projected in ImageJ. All images within a panel were collected using the same exposure times. Adjustments to levels and/or contrast for optimal viewing were applied equally to images within each panel. Images used for quantification were not processed.

## Calcium imaging

Imaging of spontaneous calcium dynamics in AWC was performed essentially as previously described (*Biron et al., 2008*). Briefly, L2 larvae were glued to an NGM agar pad on a cover glass, bathed in M9, and mounted under a second cover glass for imaging. The edge of the sandwiched cover glasses was sealed with a mixture of paraffin wax (Fisher Scientific, Pittsburgh, PA), and Vaseline, and the sample was transferred to a slide placed on a Peltier device on the microscope stage. The elapsed time from removal of the animal from the incubator to initiation of imaging was <3 min. The temperature was maintained at 20°C via temperature-regulated feedback using LabView (National Instruments, Austin, TX) and measured using a T-type thermocouple. Individual animals were imaged for 90 s at a rate of 2 Hz. Images were captured using MetaMorph (Molecular Devices, Sunnyvale, CA) and a Hamamatsu Orca digital camera. Data were analyzed using custom scripts in MATLAB (The Mathworks, Natick, MA) (*Source code 1–4*). A neuronal response was defined as the percent change of the relative fluorescence of the neuron from its baseline fluorescence level after background subtraction. A fluorescence change of >5% in AWC was considered a response. The

duration of calcium events was calculated as the sum of all events in each animal, and averaged for each genotype and condition.

Odor-evoked imaging was performed as described previously (*Jang et al., 2012*; *Ryan et al., 2014*) using custom microfluidics devices. Imaging was conducted on an Olympus BX52WI or Carl Zeiss Axio Observer A1 microscope equipped with a 40X oil objective and a Hamamatsu Orca CCD or a Zeiss Axiocam 506 mono camera. Animals were exposed to a 1 min pulse of diluted odorant in S-basal. Recording was performed at 4 Hz during the last 10 s of the pulse and 50 s following removal of the stimulus. Starved worms were transferred to an assay plate with no food before imaging. Recorded image stacks were aligned using the StackReg plugin (*Thevenaz et al., 1998*) for ImageJ (rigid body option) and were cropped to a region containing the AWC cell body. Relative changes in fluorescence, following background subtraction, were calculated using custom MATLAB scripts (*Source code 5*, *6*). Individual traces were normalized to their average baseline value for the five seconds prior to odorant removal.

## Acknowledgements

This work was supported in part by the NSF (IOS 1256488—PS), the NIH (GM 081639 and GM 103770 - PS, F32 DC013711 and T32 NS007292—MPO'D), the Human Frontiers Science Program (RGY0042/2010—PS and RAB), the DGIST R&D Program of the Ministry of Science, ICT and Future Planning (15-BD-06—KK), the National Research Foundation of Korea (NRF-2012R1A1A2009385—KK), and the T.J. Park Science Fellowship of POSCO T.J. Park Foundation (KK).

## Additional information

### Funding

| Funder | Grant reference number | Author |
|---|---|---|
| NIH Office of the Director (OD) | F32 DC013711 | Michael P O'Donnell |
| National Institutes of Health (NIH) | T32 NS007292 | Michael P O'Donnell |
| Human Frontier Science Program (HFSP) | RGY0042/2010 | Rebecca A Butcher Piali Sengupta |
| DGIST R&D Programn of the Ministry of Science, ICT and Future Planning (MSIP) | 15-BD-06 | Kyuhyung Kim |
| POSCO T.J. Park Foundation | TJ Park Science Fellowship | Kyuhyung Kim |
| National Research Foundation of Korea | 2012R1A1A2009385 | Kyuhyung Kim |
| National Science Foundation | IOS 1256488 | Piali Sengupta |
| National Institutes of Health | GM 081639 | Piali Sengupta |
| National Institutes of Health | GM 103770 | Piali Sengupta |

The funders had no role in study design, data collection and interpretation, or the decision to submit the work for publication.

### Author contributions

SJN, Conception and design, Acquisition of data, Analysis and interpretation of data, Drafting or revising the article; AT, Acquisition of data, Analysis and interpretation of data; JSP, Acquisition of data, Analysis and interpretation of data; MH, Acquisition of data, Analysis and interpretation of data; MPO, MPO'D, Analysis and interpretation of data, Drafting or revising the article; KK, Analysis and interpretation of data, Drafting or revising the article; RAB, Contributed unpublished essential data or reagents; PS, Conception and design, Analysis and interpretation of data, Drafting or revising the article

**Author ORCIDs**
Scott J Neal, http://orcid.org/0000-0002-1031-077X
Michael P O'Donnell, http://orcid.org/0000-0001-8313-8969

## Additional files

**Supplementary files**

• Supplementary file 1. List of strains used in this work.

• Source code 1. MATLAB script to quantify GCaMP intensity in ROI (spontaneous).

• Source code 2. MATLAB script for baseline correction.

• Source code 3. MATLAB script for peak response identification and quantification.

• Source code 4. MATLAB script for data compilation.

• Source code 5. MATLAB script to quantify GCaMP intensity in ROI (evoked).

• Source code 6. MATLAB subscript for TIFF image quantification.

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
