## [Decision Letter]

[Editors’ note: a previous version of this study was rejected after peer review, but the authors submitted for reconsideration. The first decision letter after peer review is shown below.]

Thank you for choosing to send your work entitled “Food-dependent regulation of developmental plasticity via CaMKI and neuroendocrine signaling” for consideration at *eLife*. Your full submission has been evaluated by a Senior Editor, a Reviewing Editor and two peer reviewers and the decision was reached after discussions between the reviewers. Based on our discussions and the individual reviews below, we regret to inform you that your work will not be considered at this time for publication in *eLife.*

There was an overall sense that the manuscript is an assembly of too many “loose ends” that emerge from multiple interesting observations-the dauer phenotype of cmk-1, the AWC ablation cmk-1 suppression data, the correlation of apparent increased basal neuronal activity of AWC in starved WT and fed cmk-1 mutants, cell-autonomous activity of cmk-1 in ASI and possible cell-non-autonomous regulation of insulin expression in ASJ. These observations did not coalesce into a coherent picture with a clear take-home message of broad interest. As you can see in the reviews, a number of suggestions were made for additional experiments. If additional data can shed further light on the phenomenon, we do encourage you to submit the manuscript to *eLife* again, but it would be a new submission.

Reviewer #1:

The authors use genetics and imaging to probe how *C. elegans* neural circuits integrate food signals into the dauer developmental decision.

1) I am wondering if *cmk-1* mutants might have altered sensitivity to pheromones and not just altered responses to food. Is there an additional assay that can be used to probe *cmk-1* responses to pheromones?

2) The starvation-driven responses in AWC are very interesting. These seem to be much much weaker than those initiated by odors. Do these signals have any significance? It is very interesting that *cmk-1* mutants have altered patterns. Moreover, starvation is still able to modulate responses in *cmk-1* mutants (Figure 2). This circuit is likely to be more complex.

3) The imaging responses are highly variable. I would suggest testing different odor concentrations to get a better handle on the responses.

4) Is the nuclear localization of *cmk-1* required to drive expression of insulins (*35* and *26*)?

5) Is *daf-2* acting downstream of *ins-35* and *26*? Also, where is *daf-2* required? The model predicts that *daf-2* might be required in ASJ to integrate food signals?

6) Fluorescence measurements of *daf-28-gfp* and *daf-7-gfp* are highly variable. I am wondering if the authors could use an independent assay to test the genetic interactions (perhaps quantitative PCR).

Reviewer #2:

The manuscript by Neal et al. describes the characterization of CMK-1 (calcium/calmodulin-dependent protein kinase I) and its role in regulating the dauer developmental decision in *C. elegans*. The authors find that the expression of a number of different neuroendocrine peptide signals, which are known to influence the dauer decision by modulating insulin and TGF-β signalling pathways, is attenuated in the *cmk-1* mutant. The authors use neuron-specific rescue experiments to implicate the function of *cmk-1* in two pairs of sensory neurons, AWC and ASI, in the regulation of the dauer decision. The data are presented clearly with relevant statistical analysis. The authors' experimental system provides an interesting opportunity to dissect the mechanisms of neuroendocrine physiology.

The basic narrative, highlighted in the Abstract and interpretation of the results, is that the expression of neuroendocrine peptides is attenuated in subsets of sensory neurons in the *cmk-1* mutant (*ins-26*, *ins-35* in AWC, *daf-7* in ASI, *daf-28* in ASJ). The similarity in expression changes that are also observed in wild type animals under starvation conditions lead the authors to suggest that CMK-1 functions in a food-dependent manner to regulate neuroendocrine responses. However, this basic narrative is at odds with the strong evidence provided by the authors that in the *cmk-1* mutant, there is an aberrant AWC neuronal state that gives rise to the production of pro-dauer signals. Genetic ablation of the AWC neurons suppresses the *cmk-1* dauer phenotype (while ablation of AWC, if anything, enhances dauer formation in wild type animals, under the specific assay conditions employed). This casts considerable doubt on the functional relevance of the data on the *ins*-26 and *ins*-35 peptides, and moreover, suggests that instead of relaying food-dependent signals to the *C. elegans* nervous system, CMK-1 may have an essential role in maintaining the function of AWC.

The authors show that knockdown of the proprotein convertase *bli-4* in AWC suppresses the *cmk-1* dauer phenotype, and the authors suggest that insulin signals from AWC promote dauer formation under limiting food. In addition, the authors show that the AWC neurons are more basally active and more responsive to IAA in starved animals or well-fed *cmk-1* mutants as compared to fed wild type animals (Figure 2), and subsequently conclude, “CMK-1 regulates AWC activity as a function of food, and that deregulated AWC activity in fed *cmk-1* mutants promotes inappropriate entry into the dauer stage.” This seems to get in the way of the overall narrative that CMK-1 has as key role in maintaining anti-dauer signals in response to food. Might ablation of AWC also suppress the diminished *daf-28*::GFP expression observed in the ASJ neurons of the *cmk-1* mutant? Does blocking this aberrant pattern of activity, without ablating AWC itself, suppress or enhance the *cmk-1* mutant?

If in fact the aberrant AWC activity leads to the production of pro-dauer signals, then one is not questioning that there are also real differences in the anti-dauer signals in the *cmk-1* mutant. It's just that their functional relevance, and correlating expression in the cmk-1 mutant with what is observed in starved wild type animals, may not yield an accurate picture of what CMK-1 actually does in normal physiology. That is, if the absence of CMK-1 simply gives rise to dysfunctional AWC neurons that produce increased levels of pro-dauer signals, then it is more difficult to infer, based on the genetic analysis alone, a pivotal physiological role for CMK-1 in the expression of neuroendocrine signals in response to food.

[Editors’ note: what now follows is the decision letter after the authors submitted for further consideration.]

Congratulations: we are very pleased to inform you that your article, “Feeding state-dependent regulation of developmental plasticity via CaMKI and neuroendocrine signaling”, has been accepted for publication in *eLife*. The Reviewing Editor for your submission was Oliver Hobert.

Reviewer #1:

The revised manuscript of Neal et al. has satisfactorily addressed my principal concern of the initial manuscript that in *cmk-1* mutants, an aberrant AWC state promotes dauer formation in a non-physiological manner. The authors show that wild type worms on live bacteria, in the presence of presumably high doses of pheromones (ascr#3 and 5), can form dauers, allowing them to show that the ablation of AWC can suppress dauer formation in wild type animals, and thus implying that AWC generates pro-dauer signals under physiological conditions, in a manner that appears to be particularly inducible by/sensitive to pheromone levels. The authors indicate these pro-dauer signals. They then go a step further and provide evidence that *ins-32* may even be this signal-function, in parallel to the original and overriding subject of their manuscript, which is that CMK-1 regulates both ILP expression to modulate *daf-28* levels in ASJ, as well as *daf-7* levels in ASI. Given the relatively large effects of removing *daf-7* and ASI on dauer formation, compared with the modest effects of removing *daf-28*, genetically dissecting the contribution of AWC signalling to the *cmk-1* phenotype was not straightforward, but the authors have put together a thoughtful explanation of how CMK-1 functions as part of the sensorineural response to food to modulate entry into dauer diapause. The authors show that the intriguing elevated basal activity of the AWC neurons in the *cmk-1* mutant, which partially mimic what the authors see upon starvation, essentially attenuates dauer formation; importantly this is shown in wild type animals, as the authors use the HisCl1 system to show that dauer formation is enhanced in both wild type and *cmk-1* mutant backgrounds on live food (and pheromone). I would add that the meaning of this elevated basal activity remains to be determined, but the observation is quite striking! The authors provide an accompanying text that is precise in stating what has been shown and what has not been shown, which builds a credible case that CMK-1 functions in at least two different pairs of neurons to regulate a feeding state-dependent developmental decision.

Reviewer #2:

The authors use genetics and imaging to identify a role for cam kinase in distinct sensory neurons to drive a dauer developmental decision. This is a beautiful study and I only have a few comments before publication.

1) The authors use the missense allele *oy21* instead of the putative null *oy20* in most of their experiments. I would not suggest repeating the experiments, but would recommend some clarification. Perhaps, the missense is non-functional and can be identified bioinformatically.

2) I am wondering if AWA rescue of cmk-1 has any effect. The promoter the authors use in Figure 1 expresses in both AWA and ASI.

3) The authors state that *ins-26* and *ins-35* is expressed in AWC. Maybe I am not reading panel 5B correctly, but it looks like these genes are expressed by 50-70% of the animals. I would suggest that the authors discuss this issue in the text.

4) The authors show that cmk-1 is localized to the AWC nuclei after 30 minutes of starvation. However, the increase in AWC calcium occurs after 2h of starvation. I am wondering about the differences between the two assays and how this affects their model. Perhaps this could be included in the Discussion.

---

## [Author Response]

[Editors’ note: the author responses to the first round of peer review follow.]

We are now submitting a completely revised manuscript incorporating data from new results. Based on issues raised by the reviewers, we have now performed several additional experiments that have led us to further develop and refine the proposed model and manuscript. As detailed in the attached ‘Response to Reviewers’, we have now addressed almost all comments from the reviewers, have included data from many new experiments, and completely reorganized and rewritten parts of the manuscript including a new Discussion.

Reviewer #1:

*The authors use genetics and imaging to probe how* C. elegans *neural circuits integrate food signals into the dauer developmental decision*.

*1) I am wondering if* cmk-1 *mutants might have altered sensitivity to pheromones and not just altered responses to food. Is there an additional assay that can be used to probe* cmk-1 *responses to pheromones?*

The reviewer raises a valid point. As discussed in the manuscript, food and pheromone signals are weighted and integrated in the dauer decision. Since pheromone and food antagonize each other in this developmental decision, decreased food sensitivity can also be read out as higher pheromone sensitivity (since animals will form more dauers at a given pheromone concentration if their threshold for food sensitivity is increased) and vice versa.

We specifically addressed this issue in the manuscript. We and others previously have shown that the ASK neurons are required to sense ascr#2 and ascr#3 in the dauer decision (1, 2). If CMK-1 acts simply by altering the sensitivity of ASK to either ascaroside, we might expect that CMK-1 would act in ASK to regulate the dauer decision. In Figure 1, we showed that this is not the case, suggesting that CMK-1 does not act simply by modifying pheromone sensitivity at the level of sensory neuron responses.

However, to test this issue further, we performed another experiment. Although there are no other readouts for pheromone sensitivity in larvae, adult hermaphrodites have previously been shown to avoid ascr#3 via the ADL sensory neurons (3). We examined ascr#3-induced intracellular calcium dynamics in adult ADL neurons in wild-type and *cmk-1* mutants, and found that *cmk-1* mutants are actually somewhat *less* sensitive to ascr#3 than wild-type animals (Image 1). We chose to not include these data in the manuscript since we wished to keep the work focused on mechanisms of dauer development, and the relationship of the ADL ascr#3 responses in adult hermaphrodites to pheromone responses in dauer development are unclear. Thus, although we cannot completely rule out that pheromone sensitivity is increased in *cmk-1* mutants in the dauer decision, taken together with additional data presented in the manuscript suggesting that the state of fed *cmk-1* mutants is similar to that of starved wild-type animals, we conclude that CMK-1 alters sensitivity to food but not to pheromones.

Author response image 1.Responses of ADL neurons in WT and *cmk-1(oy21)* mutants to ascr#3. Panels show average calcium responses of ADL to the indicated concentrations of ascr#3. Errors are SEM. Also shown is a scatter plot of peak fluorescence amplitudes of individual neuron responses in WT and *cmk-1* mutants to ascr#3. Horizontal black bars are the median. * and *** indicate *P*<0.05 and 0.001, respectively.**DOI:**
http://dx.doi.org/10.7554/eLife.10110.032

*2) The starvation-driven responses in AWC are very interesting. These seem to be much much weaker than those initiated by odors. Do these signals have any significance? It is very interesting that* cmk-1 *mutants have altered patterns. Moreover, starvation is still able to modulate responses in* cmk-1 *mutants (*Figure 2*). This circuit is likely to be more complex*.

It is possible that the AWC neurons are more depolarized in *cmk-1* mutants or in starved wild-type animals, leading to increased basal stochastic responses. However, verification of this hypothesis will require quantification of resting membrane potential which is beyond the scope of this work. However, starvation does not further modulate *cmk-1* responses in AWC (revised Figure 6, new Figure 6—figure supplement 1; please also see response to point 3 below).

The reviewer is correct in surmising that the circuit is more complex. We have now further examined the consequences of increased neuronal activity on dauer formation. We find that suppressing activity by growing animals expressing the *Drosophila* histamine-gated chloride channel in AWC (HisCl1) (4) on histamine enhances dauer formation in both wild-type and *cmk-1* mutants (new Figure 6). This result suggests that increased activity acts in parallel with the CMK-1 pathway to antagonize dauer formation. We speculate that this increased activity upon prolonged starvation in wild-type animals is a consequence of feedback from internal state that is deregulated in *cmk-1* mutants, and that this activity also serves to integrate environmental signals on longer timescales to correctly regulate dauer formation (please see revised Figure 7, revised Discussion, and response to Reviewer 2).

*3) The imaging responses are highly variable. I would suggest testing different odor concentrations to get a better handle on the responses*.

Variability in AWC responses to odorant stimuli has been observed previously [see for example Figure S1b, S1h in (5)]. As suggested by the reviewer, we have now examined responses in AWC to the removal of two concentrations of isoamyl alcohol (new Figure 6). In addition, we examined responses to the removal of one concentration of benzaldehyde, another AWC-sensed odorant (new Figure 6—figure supplement 1). Responses remain variable regardless of the odorant or its concentration, indicating that this variability may be a feature of AWC responses. Similarly, Luo et al. recently observed variability in the responses of the ASE sensory neurons to salt concentration changes that are not dependent on the specific salt concentration tested (6). To provide a clearer description, we have now included a statistical analysis of the peak odorant response amplitudes and number of responding cells in the main figure (revised Figure 6 – replaces previous Table S1; also see new Figure 6—figure supplement 1).

*4) Is the nuclear localization of* cmk-1 *required to drive expression of insulins (*35 *and* 26*)?*

We addressed this issue by expressing CMK-1+NLS and CMK-1+NES in AWC in *cmk-1(oy21)* mutants and quantifying *ins-35*p*::yfp* expression in this neuron type under fed and starved conditions. We find that expression of CMK-1+NLS suppresses the downregulated *ins-35*p*::yfp* expression phenotype of *cmk-1(oy21)* mutants in AWC in both fed and starved conditions to a slightly greater extent than expression of CMK-1+NES (revised Figure 5). These results suggest that nuclear localization of CMK-1 is required to drive expression of this ILP gene.

*5) Is* daf-2 *acting downstream of* ins-35 *and* 26*? Also, where is* daf-2 *required? The model predicts that* daf-2 *might be required in ASJ to integrate food signals?*

This is a good question. As the reviewer is no doubt aware, mutations in *daf-2* lead to pleiotropic effects including effects on dauer formation. Determining whether DAF-2 acts downstream of *ins-26/35* in ASJ requires cell-specific knockdown of *daf-2* in ASJ. However, we have had some issues (no knockdown, non-specific effects) knocking down a subset of genes in a neuron-specific manner. We are confident of the data presented in current Figure 5—figure supplement 2 since we observed effects with *bli-4* but not *egl-3* knockdown, and moreover, we observed effects upon *bli-4(RNAi)* using multiple promoters. Our model also does not require that INS-26/35 act directly on ASJ; these ILPs could act via intermediate neurons and other signaling pathways. We have now made this point clear in the model presented in revised Figure 7. We also expect that multiple ILPs may signal to ASJ, complicating interpretation of the dauer phenotypes of ASJ-specific *daf-2* knockdown. Given these caveats, at this time, we would prefer to address the site of INS-26/35 action in future experiments.

*6) Fluorescence measurements of* daf-28-gfp *and* daf-7-gfp *are highly variable. I am wondering if the authors could use an independent assay to test the genetic interactions (perhaps quantitative PCR)*.

We cannot easily measure expression differences via qPCR from whole animals since *daf-28* is expressed in at least two neuron types, and effects are observed only in ASJ in *cmk-1* mutants. Variability in *daf-7* expression has been reported previously (7, 8), and this variability is thought to be one component of the neural code reporting food abundance (7). However, to address this very valid point, we performed single molecule FISH (smFISH) to examine *daf-7* expression in wild-type and *cmk-1* mutants. As shown in Figure 2 here, we find that expression of *daf-7* remains variable, suggesting that consistent with previous observations, this variability in expression is a feature of the system. Thus, we have opted to retain the fluorescent reporter measurements in the manuscript.

Author response image 2.Expression of *daf-7* in ASI analyzed via smFISH. A) Maximum and mean pixel intensity of *daf-7* expression in ASI neurons. Median is indicated by red bars. B) Representative images showing the range of *daf-7* smFISH signals in ASI. Yellow arrowheads point to ASI cell bodies which are magnified in panels at right. Anterior is at left.Wild-type L1 larvae were fixed and hybridized with a *daf-7* probe as described previously (8). Fluorescence intensities were quantified from maximum projection images and background subtracted. n=19 neurons; 10 animals.**DOI:**
http://dx.doi.org/10.7554/eLife.10110.033

Reviewer #2:

*[…] The basic narrative, highlighted in the Abstract and interpretation of the results, is that the expression of neuroendocrine peptides is attenuated in subsets of sensory neurons in the cmk-1 mutant (*ins-26, ins-35 *in AWC,* daf*-7 in ASI,* daf*-28 in ASJ). The similarity in expression changes that are also observed in wild type animals under starvation conditions lead the authors to suggest that CMK-1 functions in a food-dependent manner to regulate neuroendocrine responses. However, this basic narrative is at odds with the strong evidence provided by the authors that in the* cmk-1 *mutant, there is an aberrant AWC neuronal state that gives rise to the production of pro-dauer signals. Genetic ablation of the AWC neurons suppresses the* cmk-1 *dauer phenotype (while ablation of AWC, if anything, enhances dauer formation in wild type animals, under the specific assay conditions employed). This casts considerable doubt on the functional relevance of the data on the* ins*-26 and* ins*-35 peptides, and moreover, suggests that instead of relaying food-dependent signals to the* C. elegans *nervous system, CMK-1 may have an essential role in maintaining the function of AWC*.

*The authors show that knockdown of the proprotein convertase* bli-4 *in AWC suppresses the* cmk-1 *dauer phenotype, and the authors suggest that insulin signals from AWC promote dauer formation under limiting food. In addition, the authors show that the AWC neurons are more basally active and more responsive to IAA in starved animals or well-fed* cmk-1 *mutants as compared to fed wild type animals (*Figure 2*), and subsequently conclude, “CMK-1 regulates AWC activity as a function of food, and that deregulated AWC activity in fed* cmk-1 *mutants promotes inappropriate entry into the dauer stage.” This seems to get in the way of the overall narrative that CMK-1 has as key role in maintaining anti-dauer signals in response to food. Might ablation of AWC also suppress the diminished* daf*-*28*::GFP expression observed in the ASJ neurons of the* cmk-1 *mutant? Does blocking this aberrant pattern of activity, without ablating AWC itself, suppress or enhance the* cmk-1 *mutant?*

*If in fact the aberrant AWC activity leads to the production of pro-dauer signals, then one is not questioning that there are also real differences in the anti-dauer signals in the* cmk-1 *mutant. It's just that their functional relevance, and correlating expression in the* cmk-1 *mutant with what is observed in starved wild type animals, may not yield an accurate picture of what CMK-1 actually does in normal physiology. That is, if the absence of CMK-1 simply gives rise to dysfunctional AWC neurons that produce increased levels of pro-dauer signals, then it is more difficult to infer, based on the genetic analysis alone, a pivotal physiological role for CMK-1 in the expression of neuroendocrine signals in response to food*.

The insightful points raised by the reviewer prompted us to perform a number of additional experiments to clarify our model and, we hope, to present a more nuanced and developed manuscript. If we may paraphrase the reviewer’s comments, the major concern is that despite the overall similarities in phenotypes between starved wild-type and fed *cmk-1* mutants suggesting a role for CMK-1 in maintaining anti-dauer signals from AWC in response to food, AWC may instead simply be ‘dysfunctional’ in *cmk-1* mutants, resulting in the production of an aberrant pro-dauer signal. In particular, the fact that AWC ablation increases dauer formation in wild-type (suggesting the presence of an anti-dauer signal), whereas AWC ablation suppresses dauer formation in *cmk-1* mutants (suggesting a possibly aberrant pro-dauer signal) is of concern. As a consequence the precise role of CMK-1 in the regulation of food-dependent neuroendocrine signals is unclear.

In brief, based on the reviewer’s points, we set out to determine whether AWC sends pro- and anti-dauer signals in both wild-type and *cmk-1* mutants as a function of feeding state, and whether it is this balance between these antagonistic signals that is disrupted in *cmk-1* mutants, as opposed to the *cmk-1* phenotypes arising from an aberrant state that may not provide accurate information about the wild-type condition. We also further analysed the role of the increased basal activity observed in AWC in *cmk-1* mutants.

a) We previously suggested that *ins-26/35* represent the anti-dauer signal from AWC based in part on the observed downregulation of expression of both genes in AWC in starved wild-type or fed *cmk-1* mutants. Based on this hypothesis, we would predict that:

i) *ins-26/35* mutants should form dauers under conditions in which wild-type animals do not

ii) loss of *ins-26/35* in the *cmk-1* mutant background should not further enhance dauer formation (since both genes are already downregulated in *cmk-1* mutants)

iii) overexpression of *ins-26/35* from AWC should rescue the enhanced dauer formation phenotype and restore *daf-28*p*::gfp* expression in ASJ in *cmk-1* mutants.

We previously presented data supporting *ii)* and *iii)* (current Figure 5). We now also show that *ins-26/35* double mutants do indeed increase dauer formation (new Figure 5). We also show that CMK-1+NLS promotes *ins-35* expression in AWC in both the fed and starved state (revised Figure 5), and rescues dauer formation to a greater extent than expression of CMK-1+NES (current Figure 4). Together, these results support the hypothesis that *ins-26/35* represent the anti-dauer signal in AWC that is downregulated upon starvation in wild-type, and inappropriately down-regulated in fed *cmk-1* mutants.

b) From the above model, we would then predict that ablation of AWC should increase dauer formation in wild-type animals (since anti-dauer signals would be abolished), but that this ablation in *cmk-1* mutants should have no further effect on dauer formation (since the anti-dauer signals are already downregulated in *cmk-1* mutants). As we reported previously, we observed that dauer formation is indeed enhanced in wild-type animals grown on heat-killed food (current Figure 5). However, dauer formation is instead suppressed in *cmk-1* mutants grown on live food (current Figure 5), indicating that AWC also produces a pro-dauer signal. (As requested by the reviewer, we also now show that ablation of AWC suppresses the diminished *daf-28*p*::gfp* expression in ASJ phenotype of *cmk-1* mutants.New Figure 5—figure supplement 1).

The reviewer raised the valid question of whether this pro-dauer signal is only observed as an aberrant feature of the *cmk-1* mutation, or whether this signal is also present in wild-type animals. We realized that wild-type AWC neurons could also send a pro-dauer signal, but this signal may be masked by the different food conditions (heat-killed vs. live food) employed to examine dauer formation in wild-type and *cmk-1* mutants, respectively. Dauer assays for *cmk-1* mutants are performed on live food since they rapidly exit dauer on heat-killed food. Wild-type animals do not generally form dauers on live food allowing us to demonstrate the shifted food sensitivity threshold for *cmk-1* mutants. However, live vs. heat-killed food provide different signals which may be sensed and integrated by distinct networks in the dauer decision, potentially confounding comparisons between wild-type and *cmk-1* mutants.

We have now succeeded in establishing conditions in which a small number of wild-type animals form dauers on live food. As now shown in new Figure 5, we find that under these conditions, ablation of AWC does indeed suppress dauer formation in wild-type animals. Consistent with this result, we also find that as in *cmk-1* mutants, ablation of AWC increases *daf-28*p*::gfp* expression in ASJ in wild-type animals grown on live food (new Figure 5—figure supplement 1). Together, these observations indicate that AWC sends both a pro- and anti-dauer signal even in wild-type animals, with the balance between these signals being regulated by food and feeding state, and that this feeding state-dependent balance is disrupted in *cmk-1* mutants. Please see the rewritten Discussion for further analysis.

c) We have now also identified a potential candidate for this pro-dauer signal. We noted previously that this pro-dauer candidate was likely to be a target of the BLI-4 proprotein convertase. Of the 19 predicted BLI-4 ILP targets (10), only *ins-32* was shown to be expressed in AWC in adult hermaphrodites (11). We have now shown (new Figure 5), that mutations in *ins-32* not only suppress the enhanced dauer formation phenotype of *cmk-1* mutants, but that they also inhibit dauer formation in wild-type animals on both live and heat-killed food. These new results imply that *ins-32* is a candidate for the pro-dauer signal from AWC, and further imply that a balance between CMK-1-regulated anti- and pro-dauer signals from AWC regulate dauer formation.

d) Finally, we explored the relationship between increased activity in AWC in wild-type animals upon prolonged starvation, or in fed *cmk-1* mutants, and dauer formation. As suggested by the reviewer, we asked whether suppression of AWC activity (not ablation) modifies dauer formation in wild-type and *cmk-1* mutants. If the aberrant activity in *cmk-1* mutants was causal to their increased dauer formation phenotype, we might expect that inhibition of this activity would suppress dauer formation in *cmk-1* mutants but may not affect dauer formation in wild-type animals.

We suppressed AWC activity by growing transgenic wild-type and *cmk-1(oy21)* animals expressing the *Drosophila* histamine-gated chloride (HisCl1) channel in AWC on 10 mM histamine (4). Suppression of AWC activity in either wild-type or *cmk-1* mutants resulted in *increased* dauer formation on live food in the presence of pheromone (new Figure 6). This result suggests that increased AWC activity upon prolonged starvation in wild-type animals, or in *cmk-1* mutants, antagonizes dauer formation. Moreover, since reducing activity enhances dauer formation similarly in both wild-type and *cmk-1* mutants, these results also indicate that increased activity acts in parallel with the CMK-1 pathway to limit dauer formation.

We now suggest that the increased activity in AWC is a consequence of feeding state-regulated feedback from internal state that is deregulated in *cmk-1* mutants. Given the long timescale of sensory signal integration in the dauer decision (12), we suggest that the availability of multiple pathways (CMK-1, activity) that integrate environmental cues on different timescales to regulate anti- and pro-dauer signals may be critical to correctly promote or suppress the dauer decision based on a temporally varying environment (please see revised Discussion).

To summarize, our new data indicate that AWC sends both pro- and anti-dauer signals based on environmental conditions, and the balance between these signals is altered in *cmk-1* mutants leading to inappropriate dauer formation under well-fed conditions. Thus, CMK-1 does indeed have an essential role in maintaining AWC function in the context of correct integration of food signals into the dauer decision.

Summary of changes: Based on the new results stemming from the reviewers’ comments, we have revised the entire manuscript including reorganizing the Results section, and have completely rewritten the Discussion. We have also now performed several new experiments, and in addition to new data provided in this document (Figure 1 and Figure 2), data from these experiments are included in the following new and revised Figures: Revised Figure 5 – new data added; New Figure 5; Revised Figure 5 – new data added; New Figure 5—figure supplement 1; Revised Figure 6, new Figure 6 – new data and analyses added; New Figure 6—figure supplement 1; New Figure 6; Revised Figure 7.

References

1. Park D, O'Doherty I, Somvanshi RK, Bethke A, Schroeder FC, Kumar U, et al. Interaction of structure-specific and promiscuous G-protein-coupled receptors mediates small-molecule signaling in Caenorhabditis elegans. Proc Natl Acad Sci USA. 2012;109: 9917-9922.

2. Kim K, Sato K, Shibuya M, Zeiger DM, Butcher RA, Ragains JR, et al. Two chemoreceptors mediate developmental effects of dauer pheromone in C. elegans. Science. 2009;326: 994-998.

3. Jang H, Kim K, Neal SJ, Macosko EZ, Kim D, Butcher RA, et al. Neuromodulatory state and sex specify alternative behaviors through antagonistic synaptic pathways in C. elegans. Neuron. 2012;75: 585-592.

4. Pokala N, Liu Q, Gordus A, Bargmann CI. Inducible and titratable silencing of Caenorhabditis elegans neurons in vivo with histamine-gated chloride channels. Proc Natl Acad Sci USA. 2014;111: 2770-2775.

5. Chalasani SH, Chronis N, Tsunozaki M, Gray JM, Ramot D, Goodman MB, et al. Dissecting a neural circuit for food-seeking behavior in Caenorhabditis elegans. Nature. 2007;450: 63-70.

6. Luo L, Wen Q, Ren J, Hendricks M, Gershow M, Qin Y, et al. Dynamic encoding of perception, memory, and movement in a C. elegans chemotaxis circuit. Neuron. 2014;82: 1115- 1128.

7. Entchev EV, Patel DS, Zhan M, Steele AJ, Lu H, Ch'ng Q. A gene-expression-based neural code for food abundance that modulates lifespan. Elife. 2015;4: e06259.

8. Meisel JD, Panda O, Mahanti P, Schroeder FC, Kim DH. Chemosensation of bacterial secondary metabolites modulates neuroendocrine signaling and behavior of C. elegans. Cell. 2014;159: 267-280.

9. Yu YV, Bell HW, Glauser DA, Goodman MB, Van Hooser SD, Sengupta P. CaMKI- dependent regulation of sensory gene expression mediates experience-dependent plasticity in the operating range of a thermosensory neuron. Neuron. 2014;84: 919-926.

10. Leinwand SG, Chalasani SH. Neuropeptide signaling remodels chemosensory circuit composition in Caenorhabditis elegans. Nat Neurosci. 2013;16: 1461-1467.

11. Takayama J, Faumont S, Kunitomo H, Lockery SR, Iino Y. Single-cell transcriptional analysis of taste sensory neuron pair in Caenorhabditis elegans. Nucleic Acids Res. 2010;38: 131-142.

12. Schaedel ON, Gerisch B, Antebi A, Sternberg PW. Hormonal signal amplification mediates environmental conditions during development and controls an irreversible commitment to adulthood. PLoS Biol. 2012;10: e1001306.